# On the Value of Interaction and Function Approximation in Imitation Learning

**Nived Rajaraman**
University of California, Berkeley
nived.rajaraman@berkeley.edu

**Yanjun Han**
University of California, Berkeley
yjhan@berkeley.edu

**Lin F. Yang**
University of California, Los Angeles
linyang@ee.ucla.edu

**Jingbo Liu**
University of Illinois, Urbana-Champaign
jingbol@illinois.edu

**Jiantao Jiao**
University of California, Berkeley
jiantao@eecs.berkeley.edu

**Kannan Ramchandran**
University of California, Berkeley
kannanr@eecs.berkeley.edu

## Abstract

We study the statistical guarantees for the Imitation Learning (IL) problem in episodic MDPs. [22] show an information theoretic lower bound that in the worst case, a learner which can even actively query the expert policy suffers from a suboptimality growing quadratically in the length of the horizon, $H$. We study imitation learning under the $\mu$-recoverability assumption of [27] which assumes that the difference in the $Q$-value under the expert policy across different actions in a state do not deviate beyond $\mu$ from the maximum. We show that the reduction proposed by [25] is statistically optimal: the resulting algorithm upon interacting with the MDP for $N$ episodes results in a suboptimality bound of $\widetilde{\mathcal{O}}\left(\mu|\mathcal{S}|H/N\right)$ which we show is optimal up to log-factors. In contrast, we show that any algorithm which does not interact with the MDP and uses an offline dataset of $N$ expert trajectories must incur suboptimality growing as $\gtrsim |\mathcal{S}|H^2/N$ even under the $\mu$-recoverability assumption. This establishes a clear and provable separation of the minimax rates between the active setting and the no-interaction setting. We also study IL with *linear function approximation*. When the expert plays actions according to a linear classifier of known state-action features, we use the reduction to multi-class classification to show that with high probability, the suboptimality of behavior cloning is $\widetilde{O}(dH^2/N)$ given $N$ rollouts from the optimal policy. This is optimal up to log-factors but can be improved to $\widetilde{O}(dH/N)$ if we have a linear expert with *parameter-sharing* across time steps. In contrast, when the MDP transition structure is known to the learner such as in the case of simulators, we demonstrate fundamental differences compared to the tabular setting in terms of the performance of an optimal algorithm, MIMIC-MD (Rajaraman et al. [22]) when extended to the function approximation setting. Here, we introduce a new problem called confidence set linear classification, that can be used to construct sample-efficient IL algorithms.

## 1 Introduction

In many practical sequential decision making problems it is difficult to manually design reward functions that capture the essence of carrying out the task "nicely". Furthermore, many modern-day

35th Conference on Neural Information Processing Systems (NeurIPS 2021).

reinforcement learning tasks operate in very large state and action spaces - with sparse reward feedback it is difficult to train good agents without additional feedback or supervision. This motivates the setting of Imitation Learning (IL) where the learner operates in a setting of unknown or unreliable rewards, but with an expert that provides demonstrations as to how to carry out the task in the desirable way. The work of [21] first showed that using expert demonstrations can significantly improve performance in autonomous driving applications. Imitation Learning approaches have found remarkable success in practice over the last decade since expert demonstrations are often available abundantly such as in game AI [13, 1], as well as more recently in autonomous-driving applications such as [5, 20]. In this paper, we study IL in the episodic Markov Decision Process (MDP) formalism.

**Notation:** An MDP $\mathcal{M} = (\mathcal{S}, \mathcal{A}, P, \rho, \mathbf{r})$, where $\mathcal{S}$ is the state space, $\mathcal{A}$ is the action space $P$ is the MDP transition, $\rho$ is initial state distribution and $\mathbf{r}$ is the reward function. The *value* $J_{\mathbf{r}}(\pi)$ of a policy $\pi$ is defined as the expected cumulative reward accumulated over an episode of length $H$, $J_{\mathbf{r}}(\pi) = \mathbb{E}_\pi[\sum_{t=1}^H \mathbf{r}_t(s_t, a_t)]$, where the notation $\mathbb{E}_\pi[\cdot]$ denotes expectation with respect to a random trajectory $\{(s_1, a_1), \cdots, (s_H, a_H)\}$ obtained by rolling out the policy $\pi = (\pi_1, \cdots, \pi_H)$, where the initial state $s_1$ is sampled independently from an initial state distribution $\rho(\cdot)$. Here $\pi_t$ denotes the policy at time step $t$. In the IL setting, we assume that the underlying reward function is *unknown and unobserved*. The reward function $\mathbf{r} = \{\mathbf{r}_1, \cdots, \mathbf{r}_H\}$ is assumed to be time-variant and pointwise bounded in $[0, 1]$, and the transition function $P = (P_1, \cdots, P_{H-1})$ of the MDP is also assumed to be time-variant. The simplest IL setting is the no-interaction setting [22].

**Definition 1** (IL in the no-interaction setting). *The learner is provided an offline dataset $D$ of $N$ trajectories (without rewards) drawn by independently rolling out an (unknown) expert policy $\pi^*$ through the MDP. The learner is not allowed to interact with the MDP.*

The learner's objective in IL is to construct a policy $\widehat{\pi}$ with small *suboptimality*, defined as the difference in the expert's and learner's values: $J_{\mathbf{r}}(\pi^*) - J_{\mathbf{r}}(\widehat{\pi})$. In this paper we restrict the expert policy to be deterministic and define $\pi_t^*(s)$ as the action played by the expert at time $t$ at state $s$. An IL instance refers to the tuple $(\mathcal{M}, \pi^*)$. The $Q$-function of a policy $\pi$ is defined as the expected reward-to-go, $Q_t^\pi(s, a) = \mathbb{E}_\pi[\sum_{t'=t}^H \mathbf{r}_{t'}(s_{t'}, a_{t'}) | s_t = s, a_t = a]$, and $f_t^\pi(s)$ is defined as the distribution over states induced at time $t$, by rolling out the policy $\pi$.

Since the expert policy is a collection of actions played at different states visited by the expert, a natural approach to IL is to use any classification algorithm to learn a mapping from states to actions as the learner's policy. This supervised learning approach has proved to be quite popular in practice and is known as behavior cloning (BC). [25] study BC from a theoretical point of view, and bound the suboptimality of a policy in terms of the 0-1 loss of the resulting classifier. More recently, in the tabular setting, [22] show that BC is statistically optimal in the no-interaction setting, achieving expected suboptimality $\lesssim \frac{|\mathcal{S}|H^2}{N}$. This $H^2$ dependence is known as *error-compounding* and is shown to be necessary even if expert is optimal or the learner can *actively query the expert*.

**Definition 2** (IL in the active setting). *In this setting, the learner is not provided a dataset of expert demonstrations up front. The learner can instead interact with the MDP for $N$ episodes. While interacting, the learner can query an oracle to return the expert's action $\pi_t^*(s)$ at the current state $s$.*

It begs the question as to why approaches such as DAGGER ([25]) and AGGREVATE ([26]) which actively query the expert often perform better than BC in practice and to explain this gap, additional assumptions must be imposed. To this end, we look at the minimax lower bound of [22] in the no-interaction setting. The key idea of the lower bound is to include an absorbing "bad" state in the MDP which is never visited in the expert dataset and offers no reward. Any policy which visits this state is doomed to incur a large suboptimality - in the absence of full information, the learner is forced to visit this often. The lower bound instance is pathological in the sense that even if the expert itself visits the bad state, it is never able to "recover" and return to the remaining states. Indeed in practical situations such as driving a car, experts often can recover and collect a high reward even if a mistake is made locally. [25] introduce an assumption to this effect, which we refer to as $\mu$-recoverability.

**Definition 3** ($\mu$-recoverability). *An IL instance is said to satisfy $\mu$-recoverability if for each $t \in [H]$ and $s \in \mathcal{S}$, $\mathbb{E}_{a \sim \pi_t^*(\cdot|s)} \left[ Q_t^{\pi^*}(s, a) \right] - Q_t^{\pi^*}(s, a) \leq \mu$ for all actions $a \in \mathcal{A}$. Informally, if the expert plays an "incorrect" action at any state $s$ at a single time $t$ and goes back to choosing the correct actions afterwards, the expected reward collected is less by at most $\mu$.*

Under the $\mu$-recoverability assumption, [25] show that a learner which minimizes the 0-1 loss under the *learner's own state distribution* to $\epsilon$ admits a suboptimality upper bound of $\mu H \epsilon$. However, it is

a-priori unclear how small $\epsilon$ can be made as a function of the number of the size of expert dataset / number of MDP interactions, $N$ in the no-interaction / active settings. This is a drawback of the reduction approach followed by [25, 27] since it cannot distinguish the power of learners in different interaction models. In this paper, we propose an policy in the active setting with expected 0-1 loss under the learner's own state distribution bounded by $|\mathcal{S}|/N$. This "completes" the reduction in a sense, and establishes suboptimality bounds for the active setting as an explicit function of the number of states $|\mathcal{S}|$, interactions $N$ and horizon $H$.

**Informal Theorem 1** (Formal version: Theorem 1 and 2). *In the active setting, under the $\mu$-recoverability assumption, there exists a learner $\widehat{\pi}$ which incurs expected suboptimality $\mathbb{E}\left[J(\pi^*) - J(\widehat{\pi})\right] \lesssim \mu H |\mathcal{S}|/N$. Furthermore, if $N \geq |\mathcal{S}|H$, for any learner $\widehat{\pi}$ in the active setting, there exists an MDP such that the expected suboptimality $\mathbb{E}\left[J(\pi^*) - J(\widehat{\pi})\right] \gtrsim \mu H |\mathcal{S}|/N$.*

The key challenge for a learner to minimize the 0-1 under its own state distribution is that the learner's policy changes over the course of optimization. Note however that it is possible to compute an unbiased estimate of the 0-1 loss under the learner's own state distribution by rolling out *just a single trajectory*. Thus the active sampling model plays a crucial role in this regard. idea is crucial towards constructing the learner policy discussed in Informal Theorem 1. Under the same $\mu$-recoverability assumption, we next consider learners in the no-interaction setting. In contrast to the active setting, we show that error compounding is unavoidable for no-interaction learners.

**Informal Theorem 2** (Formal version: Theorem 3). *In the no-interaction setting, for any learner $\widehat{\pi}$ there exists an IL instance which satisfies $\mu$-recoverability for $\mu \geq 1$ such that the expected suboptimality is $\mathbb{E}\left[J(\pi^*) - J(\widehat{\pi})\right] \gtrsim H^2 |\mathcal{S}|/N$.*

This is the first result to establish a clear *separation* in the statistical minimax rate of the suboptimality incurred by learners in the no-interaction setting such as BC, and learners which can interact with the MDP, such as DAGGER [27] and AGGRAVATE [26].

A common theme of the previous bounds in the tabular setting is that the suboptimality necessarily scales linearly in the number of states. In practical RL settings, state and actions spaces are often continuous, and thus additional assumptions are required to carry out efficient learning. In this paper, we study IL with function approximation, in particular in the linear-expert setting.

**Definition 4** (Linear-expert setting). *In this setting, for each $(s, a, t)$ tuple, the learner is provided a feature representation $\phi_t(s, a) \in \mathbb{R}^d$. For each $t \in [H]$ there exists an unknown vector $\theta_t^* \in \mathbb{R}^d$ such that $\forall s \in \mathcal{S}$, $\pi_t^*(s) = \arg\max_{a \in \mathcal{A}} \langle \theta^*, \phi_t(s, a) \rangle$.*

As we discuss in Remark 1, the linear-expert setting generalizes several known settings such as when the expert is an optimal policy under the linear-$Q^*$ assumption as well as the tabular setting with an optimal expert. We first establish a bound on the expected suboptimality incurred by BC.

**Informal Theorem 3** (Formal version: Theorem 4). *Under the linear-expert setting, the policy $\widehat{\pi}$ returned by BC incurs suboptimality $J(\pi^*) - J(\widehat{\pi}) \lesssim \frac{(d + \log(1/\delta)) H^2 \log(N)}{N}$ with probability $\geq 1 - \delta$.*

The presence of this error-compounding is not so surprising because the tabular setting with an optimal expert is a special case of the linear-expert setting where [22] show that error compounding is unavoidable for no-interaction learners. In order to break this $H^2$-dependence, we introduce a natural variant of the linear-expert setting known as linear-expert setting with parameter sharing.

**Definition 5** (Linear-expert with parameter sharing). *This setting is the same as the linear-expert setting (Definition 4), with the added constraint that for all $t$, $\theta_t^* = \theta^*$ is shared across time.*

Our main contribution is to show that in the linear-expert setting with parameter sharing, IL can be reduced to *sequence multi-class linear classification* where we learn linear classifiers from $\mathcal{S}^H \to \mathcal{A}^H$. The supervised learning reduction of [25] posits to learn separate classifiers from $\mathcal{S} \to \mathcal{A}$ or $\mathcal{S} \times [H] \to \mathcal{A}$: this fails to account for the shared parameter $\theta^*$ across time. While in both cases the resulting policy is an ERM classifier, the suboptimality grows quadratically in $H$ using the supervised learning reduction. In contrast, using the multi-class classification algorithm of [8], we also provide an algorithm $\widehat{\pi}$ with suboptimality *growing linearly in $H$*.

**Informal Theorem 4** (Formal version: Theorem 5). *Under the linear-expert setting with parameter sharing, there exists a learner $\widehat{\pi}$ with suboptimality $J(\pi^*) - J(\widehat{\pi}) \lesssim \frac{(d + \log(1/\delta)) H \log(N)}{N}$ with probability $\geq 1 - \delta$.*

With the additional linearity assumption on the expert, the learner can potentially infer the expert's action on states that are *not observed* in the dataset. However in the absence of transition information or the parameter sharing assumption, a learner cannot even distinguish between different actions at the remaining states, which is what leads to catastrophic error compounding. To remedy this issue, we borrow from the work of [22, 23] who study IL in the *known-transition setting* in tabular MDPs where the learner exactly knows the Markov transition kernel and the initial state distribution of the MDP. The motivation for this setting stems from autonomous driving applications where policies are often learned in a simulated environment prior to fine-tuning in the real world [9, 35] and in the simulator the rewards functions are still difficult to specify. In such settings, the state and action spaces are indeed unbounded, which makes it ideal to study through the frame of function approximation.

**Definition 6** (IL in the known-transition setting). *The learner is provided an dataset $D$ of $N$ trajectories (without rewards) drawn by independently rolling out the expert policy $\pi^*$ through the MDP. The learner also knows the MDP transition $P$ and initial state distribution $\rho$ exactly.*

In the tabular setting, the known transition setting has an interesting landscape: it is known from [22] that the quadratic-$H$ barrier can be broken - the authors propose the MIMIC-MD algorithm which achieves an expected suboptimality upper bound of $|\mathcal{S}|H^{3/2}/N$ and this dependence on the horizon is optimal [23]. The key idea is that with access to the MDP transition structure, as long as the visited states are conditioned to be observed in the dataset (so the expert's action is known), the learner can simulate artificial trajectories *according to the expert's policy* to generate more training data.

While the approach of simulating artificial trajectories in MIMIC-MD achieves the minimax optimal bounds here, a natural question to ask is whether the approach is tailored to work only in the tabular setting. Indeed in the presence of continuous state spaces, the learner may observe but a measure-$0$ subset of the state-space in the expert dataset. In spite of this, to apply the approach of simulating artificial trajectories, the learner must be able to *infer the expert's actions on a large fraction of the state-space*. To this end, in the known-transition setting, we propose a problem known as *confidence set linear classification* which extends multi-class linear classification and we prove that algorithms with small expected loss for confidence set linear classification can be used to construct policies with small suboptimality, using the approach of simulating artificial trajectories. At a high level, the objective of the learner is to not only otput a classifier, but also a set of inputs (confidence set) where the classifier *certifiably* outputs the correct label.

**Definition 7** (Confidence set linear classification). *Consider a classification problem on $X$ with input distribution $\rho_X$, output space $Y$, with features $\phi : X \times Y \to \mathbb{R}^d$ and a dataset $\mathcal{D}$ of $N$ examples drawn i.i.d. as $x_i \sim \rho_X$ and $y_i \sim h^*(x_i)$, where $h^*$ is an unknown linear multi-class classifier mapping $x \mapsto \arg\max_{y \in Y} \langle \theta^*, \phi(x, y) \rangle$. Given the dataset $\mathcal{D}$, a confidence set linear classifier returns a tuple $(\widehat{h}, \mathcal{X})$ where $\widehat{h}$ is any classifier from $X \to Y$ and $\mathcal{X} \subseteq X$ is a measurable set of inputs (known as the confidence set) such that $\forall x \in \mathcal{X}, \widehat{h}(x) = h^*(x)$. The learner's objective is to minimize the expected loss $\mathbb{E}\left[1 - \rho_X(\mathcal{X})\right]$.*

Sample-efficient confidence set linear classification algorithms can be used to construct learners with small suboptimality. We prove such a reduction in the linear-expert setting with linear rewards: here, in addition to the linear-expert setting, for each $t \in [H]$, the reward function $\mathbf{r}_t(\cdot, \cdot)$ is also constrained to be a linear function of the feature representations $\phi_t(\cdot, \cdot)$. The linear reward setting was considered previously in the known transition setting in [3]. We formally define it in Definition 8.

**Informal Theorem 5** (Formal version: Theorem 6). *Consider any algorithm Alg for confidence set linear classification and define $\mathfrak{R}_{N,d}(\rho_X, Y, \psi)$ as the expected loss (Definition 7) incurred by Alg when (i) the input distribution is $\rho_X$, (ii) features are $\psi : X \times Y \to \mathbb{R}^d$ and (iii) the learner is provided a dataset of $N$ samples (with labels from an unknown multi-class linear classifier). In the linear-expert setting with linear rewards, there exists a learner policy $\widehat{\pi}$ with expected suboptimality:*

$$\mathbb{E}\left[J(\pi^*) - J(\widehat{\pi})\right] \lesssim H^{3/2} \sqrt{\frac{d}{N} \frac{\sum_{t=1}^{H} \mathfrak{R}_{N,d}(f_t^{\pi^*}, \mathcal{A}, \phi_t)}{H}} \tag{1}$$

Informal Theorem 5 shows that it suffices to find good algorithms for confidence set linear classification and bound $\mathfrak{R}_{d,N}(\rho_X, Y, \phi)$ to carry out sample efficient IL. However, even in the case of binary output space $Y = \{0, 1\}$ and uniformly distributed features, the answer to this question is quite challenging and admits a non-standard rate.

**Informal Theorem 6** (Formal version: Theorem 7). *Consider an instance of confidence set linear classification where $Y = \{0, 1\}$, $\rho_X = \text{Unif}(\mathbb{S}^{d-1})$ and $\phi(x, 0) = -\phi(x, 1) = x/2 \in \mathbb{R}^d$. Then, for sufficiently large $N$,*

(i) *For any algorithm $\frac{d^{3/2}}{N\sqrt{\log(d)}} \lesssim \mathfrak{R}_{d,N,\mathcal{A}}(\rho_X, Y, \phi)$.*

(ii) *There exists an algorithm such that $\mathfrak{R}_{d,N,\mathcal{A}}(\rho_X, Y, \phi) \lesssim \frac{d^{3/2}\log(d)}{N}$.*

This result shows that the minimax risk for confidence set linear classification necessarily grows as $\gtrsim d^{3/2}/N$. This rate establishes a fundamental difference between function approximation and tabular settings. In the tabular setting, where the features for each state-action pair are orthogonal, the learner cannot conclude the labels at unobserved states. Thus, the minimax risk of confidence set linear classification corresponds to the expected probability mass on unobserved states, which is also known as the *missing mass* [18]. It is known from [22, Lemma A.20] that the expected missing mass is $\lesssim |\mathcal{S}|/N \equiv d/N$ under any distribution over states and binary action space. Informal Theorem 6 establishes a fundamental difference between the tabular setting and the linear-expert setting with linear rewards, in terms of the suboptimality guarantees achieved by the approach of simulating artificial trajectories.

While the upper bound in Informal Theorem 6 (ii) only applies in the special case of binary classification with uniformly distributed features, we conjecture that the minimax expected loss for confidence set linear classification is $\frac{d^{3/2}}{N}$. If this conjecture is true, then Informal Theorem 6 shows that there exists a learner $\widehat{\pi}$ in the known-transition setting with linear-expert and linear rewards such that the expected suboptimality is $\lesssim H^{3/2}d^{5/4}/N$. For sufficiently large $H$, this improves the $\widetilde{\Theta}(dH^2/N)$ of BC and achieves the optimal dependence on the horizon [23].

## 1.1 Related Work

There is a long line of history studying the IL problem, [2, 34, 24, 37, 25, 10, 11, 19, 14]. A number of algorithms target the $H^2$ error compounding issue, [12, 15, 17, 4, 36]. [25, 27, 4] study IL in the reduction framework, where they reduce the IL problem to a supervised learning problem and study the how the supervised learning error translates to the IL error. DAGGER [27], AGGREVATE [26], and AGGREVATED [30] learn policies by actively interacting with the environment and the expert during training. Going beyond the tabular setting, [3, 33] study IL in the presence of linear function approximation. [31] study IL in a setting where expert actions are not observed in the dataset. [6, 16] analyze DAGGER and dynamic regret under some regularity conditions, in comparison with the static regret reductions of [27], [7] propose a policy learning method called LOKI based on bootstrapping policy gradient methods using IL. [32] establish that many proposed algorithms for imitation learning indeed carry out some form of matching of distributions or features (namely, "moment matching" as studied in the paper). Our proposed algorithm for the linear-expert setting also lies in this framework.

## 2 IL with $\mu$-recoverability

As defined in Definition 3, the $\mu$-recoverability assumption captures the ability of an expert to recover and collect a high reward at a state even upon locally deviating from its action distribution at states. The reduction in [27, Theorem 2] shows that under $\mu$-recoverability, a learner policy $\widehat{\pi}$ which minimizes the 0-1 loss with respect to the expert's policy under the learner's own state distribution,

$$\mathcal{L}(f^{\widehat{\pi}}, \widehat{\pi}, \pi^*) \triangleq \frac{1}{H}\sum\nolimits_{t=1}^{H} \mathbb{E}_{s \sim f_t^{\widehat{\pi}}(\cdot)} \left[ \mathbb{E}_{a \sim \widehat{\pi}_t(\cdot|s)} \left[ \mathbb{1}(a \neq \pi_t^*(s)) \right] \right]. \tag{2}$$

to be less than $\epsilon$, incurs suboptimality upper bounded by $\mu H \epsilon$. However, in the active setting, it is a-priori unclear how small $\epsilon$ can be made as a function of the number of times the learner interacts with the MDP, $N$. We address this question in the following theorem.

**Theorem 1.** *In the active setting it is possible to construct a learner policy $\widehat{\pi}$ such that $\mathbb{E}\left[\mathcal{L}(f^{\widehat{\pi}}, \widehat{\pi}, \pi^*)\right] \lesssim |\mathcal{S}|/N$. Furthermore, under $\mu$-recoverability, $J(\pi^*) - \mathbb{E}[J(\widehat{\pi})] \lesssim \mu|\mathcal{S}|H/N$.*

Following the no-regret reduction of [27], it suffices for the learner to find a sequence of policies $\widehat{\pi}^1, \cdots, \widehat{\pi}^T$ such that the *online-learning regret*,

$$\frac{1}{N}\sum\nolimits_{i=1}^{N} \mathcal{L}(f^{\widehat{\pi}^i}, \widehat{\pi}^i, \pi^*) - \min_\pi \frac{1}{N}\sum\nolimits_{i=1}^{N} \mathcal{L}(f^{\widehat{\pi}^i}, \pi, \pi^*) \lesssim \frac{|\mathcal{S}|}{N}. \tag{3}$$

Then, the mixture policy $\frac{1}{N}\sum_{i=1}^{N}\widehat{\pi}^i$ satisfies $\mathcal{L}\left(f^{\widehat{\pi}},\widehat{\pi},\pi^*\right) \lesssim \frac{|\mathcal{S}|}{N}$. Note that in eq. (3), the oracle loss $\min_\pi \frac{1}{N}\sum_{i=1}^{N}\mathcal{L}(f^{\widehat{\pi}^i},\pi,\pi^*)$ is in fact 0, achieved by $\pi = \pi^*$. Suppose for each $i$, the learner rolls out a single trajectory according to $\widehat{\pi}^i$. Denoting the empirical state-visitation distribution $\widehat{f}^{\pi^i} = (\widehat{f}_1^{\widehat{\pi}^i},\cdots,\widehat{f}_H^{\widehat{\pi}^i})$, observe that, $\frac{1}{N}\sum_{i=1}^{N}\mathcal{L}(\widehat{f}^{\widehat{\pi}^i},\widehat{\pi}^i,\pi^*)$ is an unbiased estimate of $\frac{1}{N}\sum_{i=1}^{N}\mathcal{L}(f^{\widehat{\pi}^i},\widehat{\pi}^i,\pi^*)$ if $\widehat{\pi}^i$ is a measurable function the first $i-1$ rolled out trajectories (according to $\widehat{\pi}^1,\cdots,\widehat{\pi}^{i-1}$). Thus, it suffices for the learner to find a sequence of policies $\widehat{\pi}^1,\cdots,\widehat{\pi}^T$ which minimize the *empirical online-learning regret*: $\frac{1}{N}\sum_{i=1}^{N}\mathcal{L}(\widehat{f}^{\widehat{\pi}^i},\widehat{\pi}^i,\pi^*) - \min_\pi \frac{1}{N}\sum_{i=1}^{N}\mathcal{L}(\widehat{f}^{\widehat{\pi}^i},\pi,\pi^*)$ to be $\lesssim \frac{|\mathcal{S}|}{N}$. As we discuss in more detail in the Appendix, it is possible to construct a sequence of policies $\widehat{\pi}^1,\cdots,\widehat{\pi}^N$ using entropy-regularized mirror descent [29] which minimizes the empirical online-learning regret to be $\lesssim |\mathcal{S}|/N$. The resulting policy $\widehat{\pi} = \frac{1}{N}\sum_{i=1}^{N}\widehat{\pi}^i$ minimizes the expected 0-1 loss under its own state distribution to be $\lesssim |\mathcal{S}|/N$ in expectation. The guarantee on the expected suboptimality of this policy directly follows from [27, Theorem 2] under $\mu$-recoverability.

This suboptimality guarantee is optimal for any learner in the active setting. The lower bound instance essentially follows from that of [22] for the active tabular setting where if $N \geq |\mathcal{S}|H$, the expected suboptimality incurred is $\gtrsim \frac{|\mathcal{S}|H^2}{N}$. By scaling each reward by a factor of $\mu/H$, the same family of IL instances now satisfies $\mu$-recoverability and results in the lower bound for active learners.

**Theorem 2.** *In the active setting, if $N \geq |\mathcal{S}|H$, every learner $\widehat{\pi}$ incurs expected suboptimality $\mathbb{E}[J(\pi^*) - J(\widehat{\pi})] \gtrsim \min\{\mu, \mu|\mathcal{S}|H/N\}$.*

Now, under the same $\mu$-recoverability assumption, we study learners in the no-interaction setting. We prove a lower bound that in the worst case, error compounding is unavoidable for such learners.

**Theorem 3.** *For $|\mathcal{S}| \geq 3$ and $|\mathcal{A}| \geq H$, for any learner $\widehat{\pi}$, in the no-interaction setting, there exists an IL instance which satisfies $\mu$-recoverability for $\mu \geq 1$ such that the expected suboptimality incurred by the learner is lower bounded, $\mathbb{E}[J(\pi^*) - J(\widehat{\pi}(D))] \gtrsim \min\{H, |\mathcal{S}|H^2/N\}$.*

The lower bound we consider is a modification of the lower bound of [22] where the MDP is constructed to have a "bad" state in the MDP never visited by the expert. We modify the instance to add a single "recovery" action at the bad state; the instance now satisfies $\mu$-recoverability for any $\mu \geq 1$. If the number of actions are large $|\mathcal{A}| \geq H$, any no-interaction learner still fails to identify the recovery action with constant probability. In essence this reduces the instance to the lower bound of [22] and any no-interaction learner incurs an expected suboptimality $\gtrsim \min\{H, |\mathcal{S}|H^2/N\}$.

The classical reduction formulations of [25, 27] prove upper bounds for IL based on minimizing a certain surrogate objective. However the statistical rate of minimizing different surrogate objectives as a function of the number of interactions (active setting) / size of the expert dataset (no-interaction setting) is unclear. As we show here, with $\mu$-recoverability, the surrogate objective of 0-1 loss under the learner's policy can be minimized to $|\mathcal{S}|/N$ in the active setting, but this is impossible in the no-interaction setting. Going beyond the reduction formulation, we thus distinguish between the statistical power of learners under different interaction models.

## 3 Linear function approximation in the no-interaction setting

In this section, we go beyond the tabular setting and study IL in the presence of function approximation. In practical settings, state and action spaces are often continuous or unbounded and carrying out efficient IL requires imposing additional assumptions. In this section we study the linear-expert setting (Definition 4) where $\mathcal{S}$ and $\mathcal{A}$ may be unbounded, but the learner is provided a set of feature representations of state-actions, and the expert policy is constrained to be realizable by a unknown linear (in the feature representations) classifier. The linear-expert setting generalizes several known settings as we discuss in the following remark.

**Remark 1.** *The linear-expert setting (Definition 4) generalizes the linear-$Q^*$ setting with an optimal expert. Under this assumption, the optimal expert policy plays actions according to $\pi_t^*(s) = \arg\max_{a\in\mathcal{A}} Q_t^*(s,a) = \arg\max_{a\in\mathcal{A}}\langle\theta_t^*,\phi_t(s,a)\rangle$ for an unknown $\theta_t^* \in \mathbb{R}^d$. Thus the expert policy is realizable by a linear multi-class classifier. Since the tabular setting is a special case of the linear-$Q^*$ setting with $d = |\mathcal{S}||\mathcal{A}|$, with features for each $t$ chosen as the standard basis vectors in $\mathbb{R}^d$, the linear-expert setting with $d = |\mathcal{S}||\mathcal{A}|$ generalizes the tabular setting with an optimal expert.*

In the tabular setting, it is known that the expected suboptimality of behavior cloning is $O\left(|\mathcal{S}|H^2/N\right)$ in the worst case which is minimax optimal [22]. We first establish an upper bounds on the suboptimality incurred by BC in the linear-expert setting.

**Theorem 4.** *For $t = 1, \cdots, H$, denote $(D)_t$ as a collection of $N$ state-action pairs visited at time $t$ across trajectories in $D$. Consider a learner policy which trains a policy $\widehat{\pi}$ using BC as follows: for each $t = 1, \cdots, H$, the learner trains a linear multi-class classifier $\widehat{h}_t : \mathcal{S} \to \mathcal{A}$ on the dataset $(D)_t$ using the algorithm of [8] and plays the policy $\widehat{\pi}_t(s) = \widehat{h}_t(s)$. Then, in the linear-expert setting, with probability $1 - \delta$, the suboptimality of $\widehat{\pi}$ is upper bounded by $J(\pi^*) - J(\widehat{\pi}) \lesssim \frac{H^2(d+\log(1/\delta))\log(N)}{N}$.*

This result is in fact a special case of Theorem 5 where we prove guarantees in the *linear-expert setting with parameter sharing*, where the expert plays according to the same linear classifier shared across the episode. In Remark 2, we show that the linear-expert setting with parameter sharing with dimension $dH$ generalizes the linear-expert setting with dimension $d$.

**Remark 2.** *Linear-expert setting with dimension $d$ is a special case of the linear-expert setting with parameter sharing, with dimension $dH$. Define $\theta^* = (\theta_1^*, \cdots, \theta_H^*) \in \mathbb{R}^{dH}$ and $\phi_t'(s, a) = (0^d, \cdots, \phi_t(s, a), \cdots, 0^d) \in \mathbb{R}^{dH}$ where $\phi_t(s, a)$ is embedded in the coordinates $td + 1$ to $(t + 1)d$. Then, $\pi_t^*(s) = \arg\max_{a \in \mathcal{A}}\langle \theta_t^*, \phi_t(s, a)\rangle = \arg\max_{a \in \mathcal{A}}\langle \theta^*, \phi_t'(s, a)\rangle$, satisfying Definition 5.*

### 3.1 Linear-expert with parameter sharing: Reducing IL to sequence classification

In this section, we demonstrate a reduction of IL to sequence multi-class linear classification from $\mathcal{S}^H \to \mathcal{A}^H$, in contrast to BC which learns a classifier from $\mathcal{S} \to \mathcal{A}$. First note that the expert's policy can be thought of as a classifier from $\mathcal{S}^H \to \mathcal{A}^H$: for each input sequence of states $(s_1, s_2, \cdots, s_H)$ the expert "classifier" outputs the sequence of actions $(\pi_1^*(s_1), \pi_2^*(s_2), \cdots, \pi_H^*(s_H))$. The learner obtains $N$ i.i.d. trajectories from the expert which are examples in the classification training dataset, and the objective is to predict actions for each new trajectory from the expert. Define $\Theta$ as the set of linear multi-class classifiers for sequences, of the form

$$\mathcal{S}^H \ni (s_1, \cdots, s_H) \mapsto \underset{a_1, \cdots, a_H \in \mathcal{A}}{\arg\max} \left\langle \theta, \sum_{t=1}^{H} \phi_t(s_t, a_t) \right\rangle \in \mathcal{A}^H. \tag{4}$$

for $\theta \in \mathbb{R}^d$. Note that under the linear-expert assumption with parameter sharing, the expert's policy can be identified as a classifier in the family described above. At each state $s$, the expert plays the action according to $\arg\max_{a \in \mathcal{A}}\langle \theta^*, \phi_t(s, a)\rangle$ at time $t$. Summing over any sequence of states $s_1, \cdots, s_H$, the expert's policy therefore satisfies $(\pi_1^*(s_1), \cdots, \pi_H^*(s_H)) = \arg\max_{a_1, \cdots, a_H}\langle \theta^*, \sum_{t=1}^{H} \phi_t(s_t, a_t)\rangle$.

Note that classifiers of the form eq. (4) indeed correspond to meaningful (Markovian) policies which drawn actions $a_t$ from a policy which is a function of only the current state $s_t$. Indeed the map in eq. (4) is separable as $\sum_{t=1}^{H} \arg\max_{a_t \in \mathcal{A}}\langle \theta, \phi_t(s_t, a_t)\rangle$ where we carry out the optimization for each variable $a_1, \cdots, a_H$ separately. By contradiction, the action played by the classifier at any state $s_t$ at time $t$ must be $\arg\max_{a_t \in \mathcal{A}}\langle \theta, \phi_t(s_t, a_t)\rangle$ which is Markovian. Finally, we prove a bound on the suboptimality of the policy induced by $\widehat{\theta}$ by the expected 0-1 loss of $\widehat{\theta}$. The intuition is that in any trajectory where the learner's actions exactly match the expert's actions, no suboptimality is incurred. In contrast, in any trajectory where the learner plays an action different from the expert at some time, the suboptimality incurred is $\leq H$.

**Lemma 1.** *Consider any linear multi-class classifier $\widehat{\theta} : \mathcal{S}^H \to \mathcal{A}^H$ (in eq. (4)) with expected 0-1 loss, $\mathrm{E}_{\pi^*}[\mathbb{1}(\widehat{\theta}(s_1, \cdots, s_H) \neq (a_1, \cdots, a_H))] \leq \gamma$. Then, the policy $\widehat{\pi}$ corresponding to the linear classifier $\widehat{\theta}$, satisfies $J(\pi^*) - J(\widehat{\pi}) \leq H\gamma$.*

**Remark 3.** *The linear dependence on the horizon can be interpreted in a different way: with parameter sharing, the learner can aggregate information across time steps in an episode to learn a single linear classifier with improved guarantees. The amount of training data the learner has access to is effectively larger by a factor of $H$ since each trajectory provides $H$ samples of data for learning a single classifier common across time. The reduction analysis of [25] shows a suboptimality gap of $H^2\epsilon$ for learners with expected 0-1 loss under the expert state distribution upper bounded by $\epsilon$. If data can be aggregated across time to learn a single parameter, the expected 0-1 loss can be brought down by a factor of $H$, showing that it is possible to achieve a suboptimality bound with linear dependence on the length of the horizon.*

[8] provide a compression based algorithm for linear multi-class classification in the realizable setting. Indeed, invoking [8, Theorem 5], it is possible to learn a linear classifier $\widehat{\theta} \in \Theta$ such that the expected 0-1 loss of the classifier is upper bounded by $\frac{(d + \log(1/\delta)) \log(N)}{N}$ given $N$ expert trajectories. In conjunction with Lemma 1 this results in an upper bound on the suboptimality of the resulting policy.

**Theorem 5.** *Consider a learner $\widehat{\pi}$ which trains a classifier $\widehat{\theta}$ from the family in eq. (4) on the expert dataset using the compression based learner of [8], and at each time $t$ and state $s$, $\widehat{\pi}_t(s) = \arg\max_{a \in \mathcal{A}} \langle \widehat{\theta}, \phi_t(s, a) \rangle$. Under the linear-expert assumption with parameter sharing, with probability $\geq 1 - \delta$, the suboptimality of the learner's policy satisfies $J(\pi^*) - J(\widehat{\pi}) \lesssim \frac{H(d + \log(1/\delta)) \log(N)}{N}$.*

## 4 Linear function approximation under known-transition assumption

The framework of [23] shows that IL under the known-transition setting can be reduced to the problem of *uniform expert value estimation*: the problem of estimating the value of the expert policy under all reward functions. The authors show that given a uniform expert value estimator $\widetilde{J}_{\mathbf{r}}(\pi^*)$, which with probability $1 - \delta$ (over the expert dataset and external randomness) for all reward functions $\mathbf{r}$, satisfies $|J_{\mathbf{r}}(\pi^*) - \widetilde{J}_{\mathbf{r}}(\pi^*)| \leq \epsilon$, then the policy $\widehat{\pi}$ output by the following optimization problem,

$$\widehat{\pi} \leftarrow \arg\min_{\pi} \max_{\mathbf{r}} \widetilde{J}_{\mathbf{r}}(\pi) - J_{\mathbf{r}}(\pi) \tag{OPT}$$

incurs suboptimality $J(\pi^*) - J(\widehat{\pi}) \leq 2\epsilon$ with the same probability $1 - \delta$. [23] also show that this objective is a convex program and can be efficiently solved approximately in the tabular setting.

In this context, to execute the approach of simulating artificial trajectories, observe that a learner can construct a good estimate of the expert's value under some reward function $\mathbf{r}$ by decomposing it as the sum of two parts:

$$J_{\mathbf{r}}^1(\pi^*) = \mathbb{E}_{\pi^*}\left[\sum_{t=1}^{H} \mathbf{r}_t(s_t, a_t) \mathbb{1}(\mathcal{E})\right], \text{ and } J_{\mathbf{r}}^2(\pi^*) = \mathbb{E}_{\pi^*}\left[\sum_{t=1}^{H} \mathbf{r}_t(s_t, a_t) \mathbb{1}(\mathcal{E}^c)\right]. \tag{5}$$

where $\mathcal{E}$ is the event that the all the states $(s_1, \cdots, s_H)$ visited in the trajectory are observed in the expert dataset. The first term, $J_{\mathbf{r}}^1$, can be estimated to an arbitrary level of accuracy for any reward function $\mathbf{r}$ by rolling out many artificial trajectories using $\pi^*$, known at all states observed in the dataset. The remaining term, $J_{\mathbf{r}}^2$ can be tackled using a simple empirical estimate, as explained below. The event $\mathcal{E}^c$ guarantees that states in a trajectory are visited are observed in the dataset $D$ (where the expert's policy is known). Therefore, by holding out some trajectories in the dataset, the learner may carry out an empirical estimate of $J_{\mathbf{r}}^2(\pi^*)$ using these trajectories. The error in uniform value estimation precisely stems from the error incurred by the empirical estimate, which is shown to be $O(|\mathcal{S}|H^{3/2}/N)$ in [22], translating to the suboptimality of the policy $\widehat{\pi}$ in (OPT).

It is a natural question to ask whether this approach of simulating artificial trajectories can be applied when state and action spaces may be unbounded. To effectively use such an approach, the learner should be able to infer the expert's action at a large fraction of states in spite of *observing the expert's actions only on a measure-0 subset of states*. We show a that if the learner is able to identify the expert's policy on a known large measure of states (under the expert's state distribution), then the approach of simulating artificial trajectories can be employed to give a policy with small suboptimality. We establish such a reduction under the linear-expert setting with an additional assumption on the linearity of rewards which we introduce below. The linear reward setting was first introduced in [3] in the known-transition and discounted setting. Here, imposing the linear reward assumption enables the learner to construct linear estimates of the expert value function, which is otherwise not possible.

**Definition 8** (Linear reward assumption)**.** *Define $\mathcal{R}_{\mathrm{lin}}$ as the family of reward functions which take the form of an unknown linear function of the known feature representation of states. Namely $\mathcal{R}_{\mathrm{lin}} = \{\{\mathbf{r}_t(s, a) = \langle \omega_t, \phi_t(s, a) \rangle : t \in [H], s \in \mathcal{S}, a \in \mathcal{A}\} : \forall t \in [H], \omega_t \in \mathbb{R}^d, \|\omega_t\|_\infty \leq 1\}$. The features are assumed to satisfy $\|\phi_t(s, a)\|_1 \leq 1$. The linear reward assumption assumes the true reward function of the MDP belongs to $\mathcal{R}_{\mathrm{lin}}$.*

We propose an extension of MIMIC-MD to the linear-expert setting with linear rewards. The algorithm is based on identifying a set of states $\mathcal{X}_1, \cdots, \mathcal{X}_H$ on which the expert's policy is exactly known. The learner then constructs a uniform expert value estimator by simulating artificial trajectories using the expert policy conditioned on visiting only these states and uses an empirical estimate of the reward on the remaining states. The final policy is output using the minmax optimization problem in (OPT). A formal description is provided in Algorithm 1.

**Algorithm 1** MIMIC-MD under linear-expert and linear rewards assumption

1: **Input:** A dataset $D$ of $N$ expert policy rollouts; MDP transition $P$; feature representations $\{\phi_t(s,a) : t \in [H], s \in \mathcal{S}, a \in \mathcal{A}\}$; confidence set linear classification algorithm Alg.
2: Pick a uniformly random permutation of the trajectories of $D$ and assign the first $N/2$ as $D_0$ and the remaining trajectories as $D_1$.
3: **For** $t = 1, \cdots, H$: define $(\widehat{h}_t, \mathcal{X}_t(D_0))$ as the output of Alg$((D_0)_t)$
   $\triangleright$ $(D_0)_t$ are state-action pairs at time $t$ across trajectories in $D_0$.
   $\triangleright$ $\widehat{h}_t$ is a classifier from $\mathcal{S} \to \mathcal{A}$ and can be identified as a policy.
   $\triangleright$ $\mathcal{X}_t(D_0)$ captures a set of states on which expert's action is certifiably known.
4: Define event $\mathcal{E}_{D_0} = \{\forall t \in [H], \ s_t \in \mathcal{X}_t(D_0)\}$: all states in a trajectory belong to $\{\mathcal{X}_t(D_0)\}_{t=1}^{H}$.
5: Define the expert value estimator,

$$\widetilde{J}_{\mathbf{r}}(\pi^*) = \mathbb{E}_{\pi^*}\left[\sum_{t=1}^{H} \mathbf{r}_t(s_t, a_t)\mathbb{1}(\mathcal{E}_{D_0})\right] + \mathbb{E}_{\text{tr}\sim\text{Unif}(D_1)}\left[\sum_{t=1}^{H} \mathbf{r}_t(s_t, a_t)\mathbb{1}\left(\mathcal{E}_{D_0}^c\right)\right] \quad (6)$$

   $\triangleright$ The estimator is measurable: the first term can be estimated by rolling out many trajectories using the policy $(\widehat{h}_1, \cdots, \widehat{h}_t)$, equal to $(\pi_1^*, \cdots, \pi_H^*)$ under the measurable event $\mathcal{E}_{D_0}$
6: **Output:** Return $\widehat{\pi} \leftarrow \arg\min_{\pi} \max_{\mathbf{r} \in \mathcal{R}_{\text{lin}}} \widetilde{J}_{\mathbf{r}}(\pi^*) - J_{\mathbf{r}}(\pi)$.   $\triangleright$ $\mathcal{R}_{\text{lin}}$ is defined in Definition 8

---

**Theorem 6.** *The expected suboptimality of the policy $\widehat{\pi}$ returned by Algorithm 1 under the linear-expert setting with linear rewards can be upper bounded by,*

$$\mathbb{E}\left[J(\pi^*) - J(\widehat{\pi})\right] \lesssim H^{3/2}\sqrt{\frac{d}{N}\frac{\sum_{t=1}^{H}\mathbb{E}\left[\Pr_{\pi^*}(s_t \notin \mathcal{X}_t(D_0))\right]}{H}} \quad (7)$$

*Note that for each $t = 1, \cdots, H$, the probability $\Pr_{\pi^*}(s_t \notin \mathcal{X}_t(D_0))$ is the loss (as defined in Definition 7) of the confidence set linear classifier Alg$((D_0)_t)$ in Algorithm 1.*

As a consequence of Theorem 6, it suffices to upper bound the loss of the confidence set linear classification algorithm Alg. However, it is quite a challenging problem to compute the minimax risk for confidence set linear classification. Below, we discuss the case of binary classification.

### 4.1 Confidence set linear classification with binary outputs

In this section, we study confidence set linear classification when the output space $Y = \{0, 1\}$ is binary. Denote the dataset $D$ provided to the learner as $\{(x_1, y_1), \cdots, (x_n, y_n)\}$ where $y_i = 0$ if $\langle\theta^*, \phi(x, 0)\rangle \geq \langle\theta^*, \phi(x, 1)\rangle$ and 1 otherwise for some unknown $\theta^* \in \mathbb{R}^d$. For each sample $x_i$ observed in the dataset, the learner can conclude that $\langle\theta^*, \phi(x_i, 0)\rangle - \langle\theta^*, \phi(x_i, 1)\rangle$ is non-negative if $y_i = 0$ and is non-positive if $y_i = 1$. In other words, for each $x_i$, the learner can conclude that $\langle\theta^*, \phi(x, y) - \phi(x, 1-y)\rangle \geq 0$. Incorporating the information from all samples in the dataset, the learner can localize $\theta^*$ to a cone $\Theta = \{\theta \in \mathbb{R}^d : \forall(x, y) \in D, \ \langle\theta, \phi(x, y) - \phi(x, 1-y)\rangle \geq 0\}$. This cone captures the maximum amount of information the learner can discern about $\theta^*$. Indeed, every linear classifier $\theta \in \Theta$ correctly classifies *every* sample $x$ observed in the dataset as the correct label 0 or 1 observed in the dataset. Given the cone $\Theta$ which captures the uncertainty in $\theta^*$, one can construct a set of inputs $\mathcal{C}$ such that each classifier $\theta \in \Theta$ is consistent with their labelling of inputs in $\mathcal{C}$. We prove that largest such $\mathcal{C}$ can be directly constructed from the dataset $D$ as $\mathcal{C} = \mathcal{K} \cup -\mathcal{K}$, where $\mathcal{K}$ is the conical hull of the set of points $(\phi(x_i, 0) - \phi(x_i, 1))(-1)^{y_i}$ for $i = 1, \cdots, n$ (Figure 1).

**Lemma 2.** *The set $\mathcal{C} \subseteq X$ as defined above satisfies the following two properties:*

(i) *For each $x \in \mathcal{C}$, sign$(\langle\theta, \phi(x, 0) - \phi(x, 1)\rangle) =$ sign$(\langle\theta^*, \phi(x, 0) - \phi(x, 1)\rangle)$.*

(ii) *For any $x \notin \mathcal{C}$, any classifier $\widehat{h}$, there exists $\theta \in \Theta$ s.t. $\widehat{h}(x) \neq \mathbb{I}(\langle\theta, \phi(x, 0) - \phi(x, 1)\rangle < 0)$.*

Therefore, the learner can guarantee that the label was correctly predicted for any $x \in \mathcal{C}$. More importantly, Lemma 2 (ii) shows that for any classifier $\widehat{h}$, $\mathcal{C}$ is indeed the largest set of inputs for which the learner can guarantee to correctly predict the same output as the true classifier $\theta^*$. Thus, in the case of $\mathcal{A} = \{0, 1\}$, $\mathbb{E}[\rho_X(\mathcal{C}^c)]$ is the minimum expected loss of confidence set linear classification.

Next, we study the special case where the input space $X$ is the unit sphere $\mathbb{S}^{d-1}$, the distribution over inputs $\rho_X$ is uniform over $X$, and the feature $\phi(x, 0) = -\phi(x, 1) = x/2 \in \mathbb{R}^d$. Then

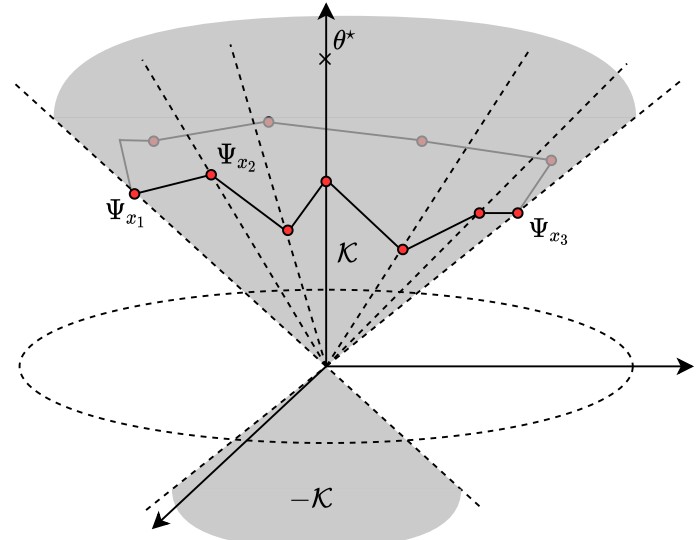

Figure 1: $\{x_1, \cdots, x_N\}$ is the set of points observed in the dataset. For each $x \in \mathbb{S}^{d-1}$, $\Psi_x \triangleq (-1)^{y_i}(\phi(x,0) - \phi(x,1))$. The gray region, $\mathcal{K}$ is the conical hull of $\{\Psi_{x_1}, \cdots, \Psi_{x_N}\}$.

$\rho_X(\mathcal{C}) = 2\rho_X(\mathcal{K})$ is the same as probability that a randomly sampled point on the surface of a hemisphere lies in the conical hull of $n$ points sampled uniformly on the surface of the hemisphere.

**Theorem 7.** *Recall that $\rho_X$ is the uniform distribution over $\mathbb{S}^{d-1}$. Then, for sufficiently large $N$,*

$$\frac{d^{3/2}}{N\sqrt{\log(d)}} \lesssim \mathbb{E}\left[\rho_X(\mathcal{C}^c)\right] \lesssim \frac{d^{3/2}\log(d)}{N}. \tag{8}$$

The proof of this result is fairly involved and we defer it to the Appendix. The key approach is to represent $\mathcal{K}$ in its dual representation and computing the probability in the dual space. The proof uses the Poissonization trick and a delicate covering argument to argue concentration in the absence of Lipschitzness. Tabular IL corresponds to the confidence set linear classification with orthogonal features. In particular, $\langle\theta, \phi(x,0) - \phi(x,1)\rangle$ is known exactly for each $x$, and for each unobserved $x$ nothing can be said about $\mathsf{sign}(\langle\theta, \phi(x,0) - \phi(x,1)\rangle)$. There, the minimax risk translates to the expected probability mass on unobserved inputs - the *missing mass* [18] which is in expectation $\lesssim d/N$ [22], which is tight over the worst-case choice of distribution on inputs, $\rho_X$. The $\widetilde{\Omega}(d^{3/2}/N)$ rate in thus Theorem 7 establishes differences between the tabular setting and the linear-expert setting with linear rewards in the context of the approach of simulating artificial trajectories.

## 5 Conclusion

We study IL in the presence of $\mu$-recoverability and under linear function approximation. In the former case, we establish a separation in the minimax expected suboptimality of learners in the no-interaction and active settings. We show upper bounds for BC under the linear expert setting and show that this quadratic dependence on $H$ can be broken in the presence of parameter sharing. Finally, we study the known transition setting, and introduce a problem known as confidence set linear classification which extends the approach of simulating artificial trajectories to the function approximation setting.

## 6 Acknowledgments and Disclosure of Funding

Nived Rajaraman and Jiantao Jiao were partially supported by NSF Grants IIS-1901252, and CCF-1909499. Part of the work was done while Lin F. Yang was visiting the Simons Institute for the Theory of Computing at UC Berkeley (the Theory of Reinforcement Learning Program). Jingbo Liu was partially supported by the starting grant from the Department of Statistics, University of Illinois. Kannan Ramchandran was supported in part by the Army Research Office (ARO) under fund 62448.

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
