# 7 Appendix A

## 7.1 Proofs for Theorem 1

**Notation:** Throughout this section, we use $\preceq$ and $\succeq$ to indicate the partial ordering on vectors where $a \preceq b$ if $a$ is not larger than $b$ co-ordinate wise ($\succeq$ is defined analogously). We also use $\mathbf{0}$ to denote the all 0's vector and $\mathbf{1}$ denote the all 1's vector (where the dimension is inferred from context). For a vector $\mathbf{0} \preceq w \in \mathbb{R}^d$, the norm $\|x\|_w^2 \triangleq \sum_{i=1}^d w_i x_i^2$ is the weighted-L2 norm.

Recall from eq. (2), the population expected 0-1 loss of a policy $\pi$ is defined as

$$\mathcal{L}(f^\pi, \pi, \pi^*) = \frac{1}{H} \sum_{t=1}^H \mathbb{E}_{S_t \sim f_t^\pi(\cdot)} \left[ \mathbb{E}_{a \sim \widehat{\pi}_t(\cdot|s)} \left[ \mathbb{1}(a \neq \pi_t^*(s)) \right] \right]. \tag{9}$$

**Lemma 3.** *Suppose there is an online learning algorithm which outputs policies $\{\widehat{\pi}^1, \cdots, \widehat{\pi}^N\}$ sequentially according to any procedure where the learner samples the policy $\widehat{\pi}_i$ from some distribution conditioned on $\mathsf{tr}_1, \cdots, \mathsf{tr}_{i-1}$, subsequently samples a trajectory $\mathsf{tr}_i$ by rolling out $\widehat{\pi}_i$, repeating this process for N iterations. Denote $\widehat{f}^{\widehat{\pi}^i} = \{\widehat{f}_1^{\widehat{\pi}^i}, \cdots, \widehat{f}_H^{\widehat{\pi}^i}\}$ where $\widehat{f}_t^{\widehat{\pi}^i}$ denotes the empirical distribution over states induced by the single trajectory $\mathsf{tr}_i$ at time t. Denote $\widehat{\pi} = \frac{1}{N} \sum_{i=1}^N \widehat{\pi}^i$ as the mixture policy. Then,*

$$\mathbb{E}\left[\mathcal{L}(f^{\widehat{\pi}}, \widehat{\pi}, \pi^*)\right] = \frac{1}{N} \sum_{i=1}^N \mathbb{E}\left[\mathcal{L}(\widehat{f}^{\widehat{\pi}^i}, \widehat{\pi}^i, \pi^*)\right]. \tag{10}$$

*Proof.* Since the trajectory $\mathsf{tr}_i$ is rolled out using $\widehat{\pi}^i$, conditioned on $\widehat{\pi}^i$, $\widehat{f}^{\widehat{\pi}^i}$ is conditionally unbiased and in expectation equal to $f^{\widehat{\pi}^i}$ (conditioned on $\widehat{\pi}^i$). Therefore, for each $i$, since $\mathcal{L}(f, \widehat{\pi}, \pi^*)$ is a linear function in $f$,

$$\mathbb{E}\left[\mathcal{L}(\widehat{f}^{\widehat{\pi}^i}, \widehat{\pi}^i, \pi^*)\Big|\widehat{\pi}^i\right] = \mathcal{L}(f^{\widehat{\pi}^i}, \widehat{\pi}^i, \pi^*). \tag{11}$$

Summing across $i = 1, \cdots, N$ and using the fact that $\mathcal{L}(f^{\widehat{\pi}}, \widehat{\pi}, \pi^*) = \frac{1}{N} \sum_{i=1}^N \mathcal{L}(f^{\widehat{\pi}^i}, \widehat{\pi}^i, \pi^*)$ and taking expectation completes the proof. $\square$

The conclusion of this lemma is that is it suffices to minimize the empirical 0-1 loss under the learner's own *one-trajectory* empirical state distribution $\frac{1}{N} \sum_{i=1}^N \mathcal{L}(\widehat{f}^{\widehat{\pi}^i}, \widehat{\pi}^i, \pi^*)$. Note that for any policy $\pi$, the loss

$$\mathcal{L}(\widehat{f}^{\widehat{\pi}^i}, \pi, \pi^*) = \frac{1}{H} \sum_{t=1}^H \sum_{s \in \mathcal{S}} \left\langle \pi_t(\cdot|s), z_t^i(s) \right\rangle \tag{12}$$

where $z_t^i(s) = \left\{ \widehat{f}_t^{\widehat{\pi}^i}(s)(1 - \pi^*(a|s)) \right\}_{a \in \mathcal{A}} \in \Delta_{\mathcal{A}}^1$ is a linear function in the policy $\pi$. The constraint on the policy variable $\pi$ is that for each $t \in [H]$ and $s \in \mathcal{S}$, $\pi_t(\cdot|s) \in \Delta_{\mathcal{A}}^1$.

Define the loss $\ell_{i,s,t}(\pi) = \sum_{s \in \mathcal{S}} \left\langle \pi_t(\cdot|s), z_t^i(s) \right\rangle$. Then the variable $\pi_t(\cdot|s)$ lies in the simplex $\Delta_{\mathcal{A}}^1$ and the vector $z_t^i(s)$ is co-ordinate wise $\geq 0$ and $\leq 1$.

To learn the sequence of policies returned by the learner, we use the normalized-EG algorithm of [28] which is also known as Follow-the-regularized-leader / Online Mirror Descent with entropy regularization for online learning. Formally, the online learning problem and the algorithm are as defined in Section 2 of [28].

**Theorem 8** (Adapted from Theorem 2.22 in [28]). *Assume that the normalized EG algorithm is run on a sequence of linear loss functions $\{\langle z_i, \cdot \rangle : i = 1, \cdots, T\}$, with $\eta = 1/2$ to return a sequence of distributions $w_1, \cdots, w_T \in \Delta_{\mathcal{A}}^1$. Assume that for all $t \in [H]$, $\mathbf{0} \preceq z_t \preceq \mathbf{1}$. For any $u$ such that $\sum_{i=1}^T \langle z_i, u \rangle = 0$,*

$$\sum_{t=1}^T \langle w_i - u, z_i \rangle \leq 4 \log(|\mathcal{A}|). \tag{13}$$

This result is adapted from Theorem 2.22 in [28] by invoking the condition that $\mathbf{0} \preceq z_t \preceq \mathbf{1}$, so the local norm $\|z_t\|_{w_t}^2$ can be upper bounded by $\langle z_t, w_t \rangle$. Choosing $\eta = \frac{1}{2}$, using the assumption that $\sum_{i=1}^{T} \langle z_i, u \rangle = 0$ and simplifying results in the statement of Theorem 8.

Suppose the learner returns the sequence of policies $\widehat{\pi}^i, \cdots, \widehat{\pi}^N$ by running the normalized EG algorithm on the sequence of losses $\ell_{1,s,t}, \cdots, \ell_{N,s,t}$ for each $s \in \mathcal{S}$ and $t \in [H]$ to return a sequence of distributions $\widehat{\pi}_t^1(\cdot|s), \cdots, \widehat{\pi}_t^N(\cdot|s) \in \Delta_{\mathcal{A}}^1$. Finally, for $i = 1, \cdots, N$, the learner returns the policy $\widehat{\pi}^i$ as $\{\{\widehat{\pi}_t^i(\cdot|s) : s \in \mathcal{S}\} : t \in [H]\}$.

Invoking the guarantee in Theorem 8 for the sequence of policies $\widehat{\pi}_t^1(\cdot|s), \cdots, \widehat{\pi}_t^N(\cdot|s)$ returned by a single instance of the normalized-EG algorithm,

$$\sum_{i=1}^{T} \langle z_t^i(s), \widehat{\pi}_t^i(\cdot|s) \rangle \leq 4 \log(|\mathcal{A}|) \tag{14}$$

Averaging across $t \in [H]$, summing across $s \in \mathcal{S}$ and recalling the definition of $\mathcal{L}$ in eq. (12) results in the bound,

$$\frac{1}{N} \sum_{i=1}^{N} \mathcal{L}(\widehat{f}^{\widehat{\pi}^i}, \widehat{\pi}^i, \pi^*) \leq \frac{4|\mathcal{S}| \log(|\mathcal{A}|)}{N}. \tag{15}$$

Finally invoking Lemma 3 shows that the resulting sequence of policies $\widehat{\pi}^1, \cdots, \widehat{\pi}^N$ and their mixtures $\frac{1}{N} \sum_{i=1}^{N} \widehat{\pi}^i$ satisfies,

$$\mathbb{E}\left[\mathcal{L}(f^{\widehat{\pi}}, \widehat{\pi}.\pi^*)\right] \leq \frac{4|\mathcal{S}| \log(|\mathcal{A}|)}{N}. \tag{16}$$

Invoking [27, Theorem 2], under $\mu$-recoverability shows that the resulting policy $\widehat{\pi}$ satisfies,

$$\mathbb{E}[J(\pi^*) - J(\widehat{\pi})] \leq \frac{4|\mathcal{S}| \log(|\mathcal{A}|)}{N}. \tag{17}$$

This completes the proof of Theorem 1.

## 7.2 Proof of Theorem 2

**Theorem 9** (Theorem 6.1 in [22])**.** *For any learner $\widehat{\pi}$, there exists an MDP $\mathcal{M}$ and a deterministic expert policy $\pi^*$ such that the expected suboptimality of the learner is lower bounded in the active setting by, $J_{\mathcal{M}}(\pi^*) - \mathbb{E}[J_{\mathcal{M}}(\widehat{\pi})] \gtrsim \min\left\{H, \frac{|\mathcal{S}|H^2}{N}\right\}$.*

For each active learner $\widehat{\pi}$ and the worst-case IL instance $(\pi^*, \mathcal{M})$ from Theorem 9, consider the IL instance $(\pi^*, \mathcal{M}_\mu)$ where the only difference between $\mathcal{M}$ and $\mathcal{M}_\mu$ is that each reward is scaled by a factor of $\mu/H \leq 1$. Note that $\mathcal{M}_\mu$ satisfies $\mu$-recoverability. Indeed, consider any state $s$. Since the rewards in $\mathcal{M}_\mu$ are in the interval $[0, \mu/H]$, the total reward of any trajectory in $\mathcal{M}_\mu$ lies in the interval $[0, \mu]$. Therefore, trivially, for each $(s, a, t) \in \mathcal{S} \times \mathcal{A} \times [H]$ tuple, $Q_t^{\pi^*}(s, \pi_t^*(s)) - Q_t^{\pi^*}(s, a) \leq \mu - 0 = \mu$ and the IL instance satisfies $\mu$-recoverability. More importantly the suboptimality of $\widehat{\pi}$ on the IL instance $(\pi^*, \mathcal{M}_\mu)$ is $\frac{\mu}{H}$ times the suboptimality under $(\pi^*, \mathcal{M})$. In other words,

$$\mathbb{E}\left[J_{\mathcal{M}_\mu}(\pi^*) - J_{\mathcal{M}_\mu}(\widehat{\pi})\right] = \frac{\mu}{H} \mathbb{E}\left[J_{\mathcal{M}}(\pi^*) - J_{\mathcal{M}}(\widehat{\pi})\right] \tag{18}$$

$$\gtrsim \frac{\mu}{H} \min\left\{H, \frac{|\mathcal{S}|H^2}{N}\right\} \tag{19}$$

$$= \min\left\{\mu, \frac{\mu|\mathcal{S}|H}{N}\right\}, \tag{20}$$

where the last inequality uses [22, Theorem 6.1]. This concludes the proof of Theorem 2.

## 7.3 Proof of Theorem 3

**Theorem 10.** *In the no-interaction setting, for any learner $\widehat{\pi}$ and $|\mathcal{S}| \geq 3$, there exists an MDP $\mathcal{M}$ with state space $\mathcal{S}$ and a deterministic expert policy $\pi^*$ which (i) satisfies the $\mu$-recovery assumption for any $\mu > 1$, and (ii) such that the expected suboptimality of the learner is lower bounded by,*

$$J_{\mathcal{M}}(\pi^*) - \mathbb{E}[J_{\mathcal{M}}(\widehat{\pi})] \gtrsim \min\left\{H, |\mathcal{S}|H^2/N\right\}. \tag{21}$$

In this section we discuss the proof of the lower bound in Theorem 10.

Define $\mathcal{S}_t(D)$ as the set of states oberved in at least one trajectory at time $t$ in the expert dataset $D$. In particular, the learner exactly knows the expert's policy $\pi_t^*(\cdot|s)$ at all states $s \in \mathcal{S}_t(D)$ for each $t = 1, \cdots, H$.

The expert policy is deterministic in the lower bound instances we construct. Define $\Pi_{\mathrm{mimic}}(D)$ as the family of policies which mimics the expert on the states visited in $D$. Namely,

$$\Pi_{\mathrm{mimic}}(D) \triangleq \Big\{ \pi : \forall t \in [H], s \in \mathcal{S}_t(D),\ \pi_t(\cdot|s) = \pi_t^*(\cdot|s) \Big\}, \tag{22}$$

Informally, $\Pi_{\mathrm{mimic}}(D)$ is the family of policies which are "compatible" with the expert dataset $D$.

In order to prove the lower bound on the worst-case expected suboptimality of any learner $\widehat{\pi}(D)$, it suffices to lower bound the Bayes expected suboptimality and find a joint distribution $\mathcal{P}$ over MDPs and expert policies satisfying $\mu$-recoverability, such that, $\mathbb{E}_{(\pi^*, \mathcal{M}) \sim \mathcal{P}} \Big[ J_\mathcal{M}(\pi^*) - \mathbb{E}\left[ J_\mathcal{M}(\widehat{\pi}(D)) \right] \Big] \gtrsim \min \Big\{ H, \frac{|\mathcal{S}|H^2}{N} \Big\}$.

**Construction of $\mathcal{P}$:**  First the expert's policy is sampled uniformly from $\Pi_{\mathrm{det}}$: for each $t \in [H]$ and $s \in \mathcal{S}$, $\pi_t^*(s) \sim \mathrm{Unif}(\mathcal{A})$. Conditioned on $\pi^*$, the distribution over MDPs induced by $\mathcal{P}$ is deterministic and given by the MDP $\mathcal{M}[\pi^*]$ in fig. 2. $\mathcal{M}[\pi^*]$ has a fixed initial distribution over states $\rho = \{\zeta, \cdots, \zeta, 1-(|\mathcal{S}|-2)\zeta, 0\}$ where $\zeta = \frac{1}{N+1}$. There is a special state $b \in \mathcal{S}$ in the MDP which has behavior different from the remaining states. At each state $s \in \mathcal{S}$, choosing the expert's action renews the state in the initial distribution $\rho$ providing a reward of 1 (except at state $b$ it provides a reward of 0), while every other action deterministically transitions the learner to the bad state and provides no reward. That is,

$$P_t(\cdot|s, a) = \begin{cases} \rho, & s \in \mathcal{S},\ a = \pi_t^*(s) \\ \delta_b, & \text{otherwise,} \end{cases} \tag{23}$$

and the reward function of the MDP is given by,

$$\mathbf{r}_t(s, a) = \begin{cases} 1, & s \in \mathcal{S} \setminus \{b\},\ a = \pi_t^*(s) \\ 0, & \text{otherwise.} \end{cases} \tag{24}$$

We first state a simple consequence of the construction of the MDP instances and $\mathcal{P}$. Note that the expert never visits the bad state $b$ by virtue of the distribution $\rho$ placing no mass on $b$. Therefore, the value of $\pi^*$ on the MDP $\mathcal{M}[\pi^*]$ is $H$.

**Lemma 4.** *The value of $\pi^*$ on the MDP $\mathcal{M}[\pi^*]$ is $H$. Namely $J_{\mathcal{M}[\pi^*]}(\pi^*) = H$.*

*Proof.* Playing the expert's action at any state in $\mathcal{S}$ is the only way to accrue non-zero reward, and in fact accrues a reward of 1. Thus the expert collects a reward of 1 at each time in any trajectory. $\square$

At the states unvisited in the dataset $D$, the learner cannot infer the expert's policy or even the transitions induced under different actions. Intuitively, the learner cannot guess the expert's action with probability $\geq 1/|\mathcal{A}|$ at such states, a statement which we prove by leveraging the Bayesian construction. In turn, the learner is forced to visit the bad state $b$ at the next point in the episode. Since the bad state is never observed in the dataset, the learner is forced to guess the expert's action to be able to recover in the distribution $\rho$ over the remaining states (lest it collects a reward of 0 for the rest of the episode). However by making $|\mathcal{A}|$ large ($\gtrsim H$), any learner, with constant probability fails to guess the expert's action at $b$ at at least a constant fraction of the episode.

Using [22, Lemma A.14], the conditional distribution of the expert's policy given the expert dataset $D$ can be characterized.

**Lemma 5.** *([22, Lemma A.14]) Conditioned on the dataset $D$ collected by the learner, the expert's deterministic policy $\pi^*$ is distributed $\sim \mathrm{Unif}(\Pi_{\mathrm{mimic}}(D))$. In other words, at each state visited in the expert dataset, the expert's choice of action is fixed as the one returned when the expert was queried at this state. At the remaining states, the expert's choice of action is sampled uniformly from $\mathcal{A}$.*

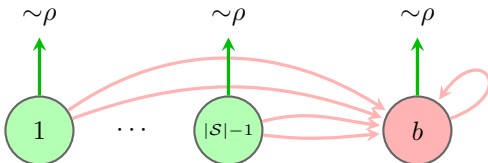

Figure 2: Upon playing the expert's (green) action at any state, the learner is renewed in the initial distribution $\rho = \{\zeta, \cdots, \zeta, 1-(|\mathcal{S}|-2)\zeta, 0\}$ where $\zeta = \frac{1}{N+1}$. Any other choice of action (red) deterministically transitions the learner to $b$.

**Definition 9.** *Define $\mathcal{P}(D)$ as the joint distribution of $(\pi^*, \mathcal{M})$ conditioned on the expert dataset $D$. Conditionally, $\pi^* \sim \mathrm{Unif}(\Pi_{\mathrm{mimic}}(D))$ and $\mathcal{M} = \mathcal{M}[\pi^*]$.*

From Lemma 5 and the definition of $\mathcal{P}(D)$ in Definition 9, applying Fubini's theorem gives,

$$\mathbb{E}_{(\pi^*, \mathcal{M}) \sim \mathcal{P}}\left[ J_{\mathcal{M}}(\pi^*) - \mathbb{E}\left[J_{\mathcal{M}}(\widehat{\pi})\right] \right] = \mathbb{E}\left[\mathbb{E}_{(\pi^*, \mathcal{M}) \sim \mathcal{P}(D)}\left[H - J_{\mathcal{M}}(\widehat{\pi}(D))\right]\right]. \tag{25}$$

Next we relate this to the first time the learner visits a state unobserved in $D$.

**Lemma 6.** *Define the stopping time $\tau$ as the first time $t$ that the learner encounters a state $s_t \neq b$ that has not been visited in $D$ at time $t$. That is,*

$$\tau = \begin{cases} \inf\{t : s_t \notin \mathcal{S}_t(D) \cup \{b\}\} & \exists t : s_t \notin \mathcal{S}_t(D) \cup \{b\} \\ H + 1 & \text{otherwise}. \end{cases} \tag{26}$$

*Then, conditioned on the expert dataset $D$,*

$$\mathbb{E}_{(\pi^*, \mathcal{M}) \sim \mathcal{P}(D)}\left[ J(\pi^*) - \mathbb{E}\left[J(\widehat{\pi})\right] \right] \geq \left(1 - \frac{1}{|\mathcal{A}|}\right)^{H+1} \mathbb{E}_{(\pi^*, \mathcal{M}) \sim \mathcal{P}(D)}\left[\mathbb{E}_{\widehat{\pi}(D)}\left[H - \tau\right]\right]. \tag{27}$$

We defer the proof of this result to the end of this section.

Plugging the result of Lemma 6 into eq. (25), and recalling the assumption that $|\mathcal{A}| \geq H + 1$,

$$\mathbb{E}_{(\pi^*, \mathcal{M}) \sim \mathcal{P}}\left[ J(\pi^*) - \mathbb{E}\left[J(\widehat{\pi})\right] \right] \geq \frac{1}{4}\mathbb{E}\left[\mathbb{E}_{(\pi^*, \mathcal{M}) \sim \mathcal{P}(D)}\left[\mathbb{E}_{\widehat{\pi}}\left[H - \tau\right]\right]\right], \tag{28}$$

$$\overset{(i)}{\geq} \frac{H}{8}\mathbb{E}\left[\mathbb{E}_{(\pi^*, \mathcal{M}) \sim \mathcal{P}(D)}\left[\mathrm{Pr}_{\widehat{\pi}}\left[\tau \leq \lfloor H/2 \rfloor\right]\right]\right], \tag{29}$$

$$= \frac{H}{8}\mathbb{E}_{(\pi^*, \mathcal{M}) \sim \mathcal{P}}\left[\mathbb{E}\left[\mathrm{Pr}_{\widehat{\pi}}\left[\tau \leq \lfloor H/2 \rfloor\right]\right]\right], \tag{30}$$

where $(i)$ uses Markov's inequality, and the last equation uses Fubini's theorem.

The last remaining element of he proof is to indeed bound the probability that the learner visits a state unobserved in the dataset before time $\lfloor H/2 \rfloor$ which immediately follows from [22, Lemma A.16] shows that for any learner $\widehat{\pi}$, $\mathbb{E}_{(\pi^*, \mathcal{M}) \sim \mathcal{P}}\left[\mathbb{E}\left[\mathrm{Pr}_{\widehat{\pi}}\left[\tau \leq \lfloor H/2 \rfloor\right]\right]\right]$ is lower bounded by $\gtrsim \min\{1, |\mathcal{S}|H/N\}$. Therefore,

$$\mathbb{E}_{(\pi^*, \mathcal{M}) \sim \mathcal{P}}\left[ J(\pi^*) - \mathbb{E}\left[J(\widehat{\pi})\right] \right] \gtrsim H \min\left\{1, \frac{|\mathcal{S}|H}{N}\right\}. \tag{31}$$

as long as $|\mathcal{A}| \geq H + 1$.

**Lemma 7.** *([22, Lemma A.16]) For any learner policy $\widehat{\pi}$,*

$$\mathbb{E}_{(\pi^*, \mathcal{M}) \sim \mathcal{P}}\left[\mathbb{E}\left[\mathrm{Pr}_{\widehat{\pi}}\left[\tau \leq \lfloor H/2 \rfloor\right]\right]\right] \geq 1 - \left(1 - \frac{|\mathcal{S}| - 2}{e(N+1)}\right)^{\lfloor H/2 \rfloor} \gtrsim \min\left\{1, \frac{|\mathcal{S}|H}{N}\right\}. \tag{32}$$

Finally, we prove Lemma 6.

*Proof of Lemma 6.* Define the random time $\tau_b$ to be the first time the learner encounters the state $b$ while rolling out a trajectory. Formally,

$$\tau_b = \begin{cases} \inf\{t : s_t = b\} & \exists t : s_t = b \\ H + 1 & \text{otherwise.} \end{cases}$$

Furthermore, define $\Gamma_b$ as the random variable which counts the number of time steps the trajectory stays in the state $b$ after visiting it for the first time. Namely,

$$\Gamma_b = \begin{cases} \inf\{\Delta \geq 0 : s_{\tau_b + \Delta + 1} \neq b\} & \tau_b \leq H \\ 0 & \text{otherwise.} \end{cases} \tag{33}$$

Since the state $b$ always dispenses 0 reward and since $\mathbf{r}$ is bounded in $[0, 1]$, conditioned on the expert dataset $D$,

$$H - \mathbb{E}_{(\pi^*, \mathcal{M}) \sim \mathcal{P}(D)} [J(\widehat{\pi})] = H - \mathbb{E}_{(\pi^*, \mathcal{M}) \sim \mathcal{P}(D)} \left[ \mathbb{E}_{\widehat{\pi}} \left[ \sum_{t=1}^{H} \mathbf{r}_t(s_t, a_t) \right] \right] \tag{34}$$

$$\geq \mathbb{E}_{(\pi^*, \mathcal{M}) \sim \mathcal{P}(D)} [\mathbb{E}_{\widehat{\pi}} [\Gamma_b]] \tag{35}$$

Fixing the expert dataset $D$ and the expert's policy $\pi^*$ (which determines the MDP $\mathcal{M}[\pi^*]$), we under the distribution of $\Gamma_b$.

To this end, first observe that for any $t \leq H - 1$ and state $s \in \mathcal{S}$,

$$\Pr_{\widehat{\pi}} [\Gamma_b \geq \Delta + 1, \Gamma_b \geq \Delta, \tau_b = t] \tag{36}$$

$$= \Pr_{\widehat{\pi}} [\Gamma_b \geq \Delta + 1 | \Gamma_b \geq \Delta, \tau_b = t] \Pr_{\widehat{\pi}} [\Gamma_b \geq \Delta, \tau_b = t] \tag{37}$$

$$= \left( 1 - \widehat{\pi}_{t+\Delta}(\pi^*_{t+\Delta}(b)|b) \right) \Pr_{\widehat{\pi}} [\Gamma_b \geq \Delta, \tau_b = t]. \tag{38}$$

where in the last equation, we use the fact that the learner must play an action other than $\pi^*_{t+\Delta}(b)$ to stay in state $b$ at time $t + \Delta$. Next, we take expectation with respect to the randomness of $\pi^*$. Conditioned on $D$, $\pi^*$ is sampled uniformly from the set of policies $\Pi_{\mathrm{mimic}}(D)$ (Lemma 5). In particular, conditioned on $D$, the expert policy is sampled independently at states. Conditioned on $\pi^*$, the underlying MDP is $\mathcal{M}[\pi^*]$. Observe that the dependence of the second term $\Pr_{\widehat{\pi}} [\tau = t, s_t = s]$ on $\pi^*$ comes from the probability computed with the underlying MDP chosen as $\mathcal{M}[\pi^*]$. Observe that it only depends on the characteristics of $\mathcal{M}[\pi^*]$ till time $t - 1$ which are determined by $\pi^*_1, \cdots, \pi^*_{t+\Delta-1}$. On the other hand, the first term $\left( 1 - \widehat{\pi}_t(\pi^*_{t+\Delta}(b)|b) \right)$ depends only on the random variable $\pi^*_{t+\Delta}$. As a consequence, the two terms depend on a disjoint set of random variables which are independent.

Taking expectation with respect to the randomness of $\pi^* \sim \mathrm{Unif}(\Pi_{\mathrm{mimic}}(D))$ and $\mathcal{M} = \mathcal{M}[\pi^*]$ (which defines the joint distribution $\mathcal{P}(D)$ in eq. (22)), for $0 \leq \Delta \leq H - t$ and $t \in [H]$,

$$\mathbb{E}_{(\pi^*, \mathcal{M}) \sim \mathcal{P}(D)} \left[ \Pr_{\widehat{\pi}} [\Gamma_b \geq \Delta + 1, \Gamma_b \geq \Delta, \tau_b = t] \right]$$

$$= \mathbb{1}(t + \Delta \leq H) \cdot \mathbb{E}_{(\pi^*, \mathcal{M}) \sim \mathcal{P}(D)} \left[ 1 - \widehat{\pi}_{t+\Delta}(\pi^*_{t+\Delta}(b)|b) \right] \mathbb{E}_{(\pi^*, \mathcal{M}) \sim \mathcal{P}(D)} \left[ \Pr_{\widehat{\pi}} [\Gamma_b \geq \Delta, \tau_b = t] \right] \tag{39}$$

$$= \mathbb{1}(t + \Delta \leq H) \cdot \left( 1 - \frac{1}{|\mathcal{A}|} \right) \mathbb{E}_{(\pi^*, \mathcal{M}) \sim \mathcal{P}(D)} \left[ \Pr_{\widehat{\pi}} [\Gamma_b \geq \Delta, \tau_b = t] \right], \tag{40}$$

where in the last equation we use the fact that the state $b$ is never observed in the expert dataset. So conditioned on $D$, $\pi^*_{t+\Delta}(b)$ is sampled uniformly from $\mathcal{A}$. By upper bounding $\Pr_{\widehat{\pi}} [\Gamma_b \geq \Delta + 1, \Gamma_b \geq \Delta, \tau_b = t] \leq \Pr_{\widehat{\pi}} [\Gamma_b \geq \Delta + 1, \tau_b = t]$ results in the inequality,

$$\mathbb{E}_{(\pi^*, \mathcal{M}) \sim \mathcal{P}(D)} \left[ \Pr_{\widehat{\pi}} [\Gamma_b \geq \Delta + 1, \tau_b = t] \right]$$

$$\geq \mathbb{1}(t + \Delta \leq H) \cdot \left( 1 - \frac{1}{|\mathcal{A}|} \right) \mathbb{E}_{(\pi^*, \mathcal{M}) \sim \mathcal{P}(D)} \left[ \Pr_{\widehat{\pi}} [\Gamma_b \geq \Delta, \tau_b = t] \right] \tag{41}$$

Unrolling the equation, for each $\Delta = 0, 1, \cdots, H - t + 1$ we have,

$$\mathbb{E}_{(\pi^*, \mathcal{M}) \sim \mathcal{P}(D)} \left[ \Pr_{\widehat{\pi}} [\Gamma_b \geq \Delta, \tau_b = t] \right]$$

$$\geq \left( 1 - \frac{1}{|\mathcal{A}|} \right)^{\Delta} \mathbb{E}_{(\pi^*, \mathcal{M}) \sim \mathcal{P}(D)} \left[ \Pr_{\widehat{\pi}} [\tau_b = t] \right] \tag{42}$$

Summing up over $\Delta = 0, 1, \cdots, H - t + 1$,

$$\mathbb{E}_{(\pi^*, \mathcal{M}) \sim \mathcal{P}(D)} \Big[ \mathbb{E}_{\widehat{\pi}} \left[ \Gamma_b \mathbb{1}(\tau_b = t) \right] \Big]$$

$$\geq \sum_{\Delta=0}^{H-t+1} \left( 1 - \frac{1}{|\mathcal{A}|} \right)^{\Delta} \mathbb{E}_{(\pi^*, \mathcal{M}) \sim \mathcal{P}(D)} \Big[ \Pr_{\widehat{\pi}} \left[ \tau_b = t \right] \Big] \tag{43}$$

$$\geq (H - t + 1) \left( 1 - \frac{1}{|\mathcal{A}|} \right)^{H} \mathbb{E}_{(\pi^*, \mathcal{M}) \sim \mathcal{P}(D)} \Big[ \Pr_{\widehat{\pi}} \left[ \tau_b = t \right] \Big]. \tag{44}$$

Finally summing across $t = 1, \cdots, H + 1$,

$$\mathbb{E}_{(\pi^*, \mathcal{M}) \sim \mathcal{P}(D)} \Big[ \mathbb{E}_{\widehat{\pi}} \left[ \Gamma_b \right] \Big] \geq \left( 1 - \frac{1}{|\mathcal{A}|} \right)^{H} \mathbb{E}_{(\pi^*, \mathcal{M}) \sim \mathcal{P}(D)} \left[ \mathbb{E}_{\widehat{\pi}} \left[ H - \tau_b + 1 \right] \right]. \tag{45}$$

Finally, we invoke [22, Lemma A.15] and in particular, eq. (134) to arrive at the desired bound.

$$\mathbb{E}_{(\pi^*, \mathcal{M}) \sim \mathcal{P}(D)} \Big[ \mathbb{E}_{\widehat{\pi}} \left[ \Gamma_b \right] \Big] \geq \left( 1 - \frac{1}{|\mathcal{A}|} \right)^{H+1} \mathbb{E}_{(\pi^*, \mathcal{M}) \sim \mathcal{P}(D)} \left[ \mathbb{E}_{\widehat{\pi}} \left[ H - \tau \right] \right] \tag{46}$$

Note that although the MDP family considered in [22, Lemma A.15] is different, until the state $b$ is visited the two MDPs are identical and therefore $\tau$ and $\tau_b$ are distributed identically under either MDP family for the same policy. □

### 7.4 Proof of Theorems 4 and 5

*Proof of Lemma 1.* Conditioned on the expert and the learner playing the same actions in the state, the error of the learner is exactly 0 since in such trajectories both policies collect the same reward. On the other hand, when the learner plays an action different from the expert at a visited state (and thus the 0-1 loss for this trajectory is 1), the maximum error the learner can incur is $H$. □

*Proof of Theorem 5.* Consider the compression based multi-class linear classification algorithm of [8]. This algorithm admits the following guarantee for multi-class sequence classification.

**Theorem 11** (Theorem 5 in [8])**.** *Consider any linear multi-class classification problem with features* $\phi : X \times Y \to \mathbb{R}$. *The learner is provided samples* $D = \{(x_1, y_1), \cdots, (x_n, y_n)\}$, *where each* $x_i$ *is sampled from an unknown distribution* $\rho$ *and with label* $y_i = \arg\max_{y \in Y} \langle \phi(x, y), \theta^* \rangle$ *for an unknown* $\theta^* \in \mathbb{R}^d$. *Then, if* $n \geq \frac{d \log(1/\epsilon) + \log(1/\delta)}{\epsilon}$, *with probability* $\geq 1 - \delta$ *the compression algorithm of [8] returns a linear function* $\widehat{\theta} \in \mathbb{R}^d$ *such that, the expected* 0-1 *loss is bounded by* $\epsilon$. *Namely,*

$$\mathbb{E}_{x \sim \rho} \left[ \mathbb{1} \left( \arg\max_{y \in Y} \left\langle \phi(x, y), \widehat{\theta} \right\rangle \neq \arg\max_{y \in Y} \langle \phi(x, y), \theta^* \rangle \right) \right] \leq \epsilon \tag{47}$$

Consider the dataset as a mapping from sequences of states to sequences of actions $\mathcal{S}^H \to \mathcal{A}^H$. In addition, the expert policy can be thought of as a classifier from $\mathcal{S}^H \to \mathcal{A}^H$. In the sense: for $\mathcal{S}^H \ni (s_1, \cdots, s_H) \mapsto (\pi_1^*(s_1), \cdots, \pi_H^*(s_H)) \in \mathcal{A}^H$. A sequence classifier is a mapping from $\mathcal{S}^H \to \mathcal{A}^H$.

A sequence linear classifier is defined as: For $\theta \in \mathbb{R}^d$, the corresponding linear sequence classifier is $(s_1, \cdots, s_H) \mapsto \arg\max_{a_1, \cdots, a_H \in \mathcal{A}} \theta \mapsto \left\langle \theta, \sum_{t=1}^{H} \phi_t(s_t, a_t) \right\rangle$. Define the set of linear sequence classifiers corresponding to $\theta \in \mathbb{R}^d$. Then, the following two propositions are true:

i. The expert policy $\pi^*$ is a linear sequence classifier under the linear-expert assumption. At any time $t$ and state $s_t$, the expert chooses the action $a_t = \arg\max_{a \in \mathcal{A}} \langle \theta_t^*, \phi_t(s, a) \rangle$. Summing across any sequence of states $(s_1, \cdots, s_H)$ shows that the sequence of actions played by the expert satisfies: $(a_1, \cdots, a_H) = \arg\max_{a_1', \cdots, a_H' \in \mathcal{A}} \langle \theta, \phi_t(s_t, a_t) \rangle$ which proves the claim.

ii. Every sequence linear classifier corresponds to a meaningful Markovian policy. Indeed, for some $\theta \in \mathbb{R}^d$, consider the sequence linear classifier corresponding to $\theta$. If at a state $s_t$ at time $t$, the classifier does not choose the action $a_t = \arg\max_{a \in \mathcal{A}} \langle \theta, \phi_t(s_t, a) \rangle$, then on any sequence that visits the state $s_t$ at time $t$, $(a_1, \cdots, a_H) \neq \arg\max_{a'_1, \cdots, a'_H \in \mathcal{A}} \langle \theta, \phi_t(s_t, a'_t) \rangle$ which leads to a contradiction. Therefore, the sequence linear classifier plays the action $a_t = \arg\max_{a \in \mathcal{A}} \langle \theta, \phi_t(s_t, a) \rangle$ at each state $s_t$ at each time $t$. It is therefore a Markovian policy.

The implication of these two points is that it suffices to find a sequence linear classification algorithm from $\mathcal{S}^H \to \mathcal{A}^H$ with small expected 0-1 error, given a dataset of trajectories from the expert policy. Invoking the algorithm of [8] for linear multi-class classification and Theorem 11 completes the proof shows that indeed there is a linear sequence classifier with expected 0-1 loss upper bounded by $\frac{(d + \log(1/\delta)) \log(N)}{N}$ which completes the proof, invoking Lemma 1.

The proof of Theorem 4 follows immediately as a corollary of Theorem 5, by invoking Remark 2.

$\square$

## 7.5 Reduction of IL to Confidence set linear classification

*Proof of Theorem 6.* By decomposing as the sum of two parts, $J(\pi^*) = \sum_{t=1}^{H} \mathbb{E}_{\pi^*} [\mathbf{r}_t(s_t, a_t) \mathbb{1}(\mathcal{E}_{D_0})] + \sum_{t=1}^{H} \mathbb{E}_{\pi^*} [\mathbf{r}_t(s_t, a_t) \mathbb{1}(\mathcal{E}_{D_0}^c)]$ and recalling from definition that $\widetilde{J}_{\mathbf{r}}(\pi^*, D) = \mathbb{E}_{\pi^*} [\sum_{t=1}^{H} \mathbf{r}_t(s_t, a_t) \mathbb{1}(\mathcal{E}_{D_0})] + \mathbb{E}_{\text{tr} \sim \text{Unif}(D_1)} [\sum_{t=1}^{H} \mathbf{r}_t(s_t, a_t) \mathbb{1}(\mathcal{E}_{D_0}^c)]$, we have that,

$$J(\pi^*) - \widetilde{J}(\pi^*, D) = \sum_{t=1}^{H} \mathbb{E}_{\pi^*} [\mathbf{r}_t(s_t, a_t) \mathbb{1}(\mathcal{E}_{D_0}^c)] - \mathbb{E}_{\text{tr} \sim \text{Unif}(D_1)} [\mathbf{r}_t(s_t, a_t) \mathbb{1}(\mathcal{E}_{D_0}^c)]. \quad (48)$$

By the linear reward assumption,

$$J(\pi^*) - \widetilde{J}(\pi^*, D) = \sum_{t=1}^{H} \langle \mathbb{E}_{\pi^*} [\phi_t(s_t, a_t) \mathbb{1}(\mathcal{E}_{D_0}^c)] - \mathbb{E}_{\text{tr} \sim \text{Unif}(D_1)} [\phi_t(s_t, a_t) \mathbb{1}(\mathcal{E}_{D_0}^c)], \theta_t \rangle \quad (49)$$

Therefore, by Holder's inequality,

$$\sup_{(\theta_1, \cdots, \theta_t): \forall t, \|\theta_t\|_\infty \leq 1} \left| J(\pi^*) - \widetilde{J}(\pi^*, D) \right| \quad (50)$$

$$\leq \sum_{t=1}^{H} \left\| \mathbb{E}_{\pi^*} [\phi_t(s_t, a_t) \mathbb{1}(\mathcal{E}_{D_0}^c)] - \mathbb{E}_{\text{tr} \sim \text{Unif}(D_1)} [\phi_t(s_t, a_t) \mathbb{1}(\mathcal{E}_{D_0}^c)] \right\|_1. \quad (51)$$

Taking expectation over $D_0$ and $D_1$,

$$\mathbb{E}_D \left[ \sup_{(\theta_1, \cdots, \theta_t): \forall t, \|\theta_t\|_\infty \leq 1} \left| J(\pi^*) - \widetilde{J}(\pi^*, D) \right| \right] \quad (52)$$

$$\leq \sum_{t=1}^{H} \mathbb{E}_{D_0} \left[ \mathbb{E}_{D_1} \left[ \left\| \mathbb{E}_{\pi^*} [\phi_t(s_t, a_t) \mathbb{1}(\mathcal{E}_{D_0}^c)] - \mathbb{E}_{\text{tr} \sim \text{Unif}(D_1)} [\phi_t(s_t, a_t) \mathbb{1}(\mathcal{E}_{D_0}^c)] \right\|_1 \Big| D_0 \right] \right] \quad (53)$$

$$\overset{(i)}{\leq} \sum_{t=1}^{H} \sum_{i=1}^{d} \mathbb{E}_{D_0} \left[ \mathbb{E}_{D_1} \left[ \left| \mathbb{E}_{\pi^*} [\langle \phi_t(s_t, a_t), e_i \rangle \mathbb{1}(\mathcal{E}_{D_0}^c)] - \mathbb{E}_{\text{tr} \sim \text{Unif}(D_1)} [\langle \phi_t(s_t, a_t), e_i \rangle \mathbb{1}(\mathcal{E}_{D_0}^c)] \right| \Big| D_0 \right] \right] \quad (54)$$

$$\overset{(ii)}{\leq} \sum_{t=1}^{H} \sum_{i=1}^{d} \mathbb{E}_{D_0} \left[ \left( \mathbb{E}_{D_1} \left[ \left( \mathbb{E}_{\pi^*} [\langle \phi_t(s_t, a_t), e_i \rangle \mathbb{1}(\mathcal{E}_{D_0}^c)] - \mathbb{E}_{\text{tr} \sim \text{Unif}(D_1)} [\langle \phi_t(s_t, a_t), e_i \rangle \mathbb{1}(\mathcal{E}_{D_0}^c)] \right)^2 \Big| D_0 \right] \right)^{1/2} \right] \quad (55)$$

where in $(i)$ $\{e_i : i = 1, \cdots, d\}$ denotes the set of standard basis vectors in $\mathbb{R}^d$ and $(ii)$ follows by Jensen's inequality. Finally, we bound the variance of the expected feature under $\pi^*$ and under the

empirical distribution in $D_1$ (which is independent of $D_0$). For each $i = 1, \cdots, d$ the variance of the co-ordinate $i$ is upper bounded by $\frac{\mathbb{E}_{\pi^*}\left[\langle \phi_t(s_t,a_t),e_i\rangle^2 \mathbb{1}(\mathcal{E}^c_{D_0})\right]}{N}$. Therefore,

$$\mathbb{E}_D\left[\sup_{(\theta_1,\cdots,\theta_t):\forall t, \|\theta_t\|_\infty \leq 1} \left|J(\pi^*) - \widetilde{J}(\pi^*, D)\right|\right] \tag{56}$$

$$\leq \sqrt{\frac{1}{N}} \sum_{t=1}^H \sum_{i=1}^d \mathbb{E}_{D_0}\left[\left(\mathbb{E}_{\pi^*}\left[\langle \phi_t(s_t,a_t),e_i\rangle^2 \mathbb{1}(\mathcal{E}^c_{D_0})\right]\right)^{1/2}\right] \tag{57}$$

$$\overset{(i)}{\leq} \sum_{t=1}^H \sqrt{\frac{d}{N}} \left(\sum_{i=1}^d \mathbb{E}_{D_0}\left[\mathbb{E}_{\pi^*}\left[|\langle \phi_t(s_t,a_t),e_i\rangle| \, \mathbb{1}(\mathcal{E}^c_{D_0})\right]\right]\right)^{1/2} \tag{58}$$

$$= \sum_{t=1}^H \sqrt{\frac{d}{N}} \left(\mathbb{E}_{D_0}\left[\mathbb{E}_{\pi^*}\left[\|\phi_t(s_t,a_t)\|_1 \mathbb{1}(\mathcal{E}^c_{D_0})\right]\right]\right)^{1/2} \tag{59}$$

$$\leq H\sqrt{\frac{d}{N}\mathbb{E}_{D_0}\left[\Pr\left(\mathcal{E}^c_{D_0}\right)\right]} \tag{60}$$

where $(i)$ follows by an application of Jensen's inequality and Cauchy Schwarz inequality and using the fact that $\langle \phi_t(s_t,a_t),e_i\rangle^2 \leq |\langle \phi_t(s_t,a_t),e_i\rangle|$ since $\|\phi_t(s_t,a_t)\|_1 \leq 1$. By union bounding, $\Pr\left(\mathcal{E}^c_{D_0}\right) \leq \sum_{t=1}^H \Pr_{\pi^*}(s_t \notin \mathcal{X}_t(D_0))$ which completes the proof of Theorem 6. $\qquad\square$

### 7.6 Proof of Lemma 2

*Proof of Lemma 2.* (i) Observe that for any input $x$, if $x \in \mathcal{K}$, then $x = \phi(x,0) - \phi(x,1)$ can almost surely be expressed as a positive linear combination $\sum_{i=1}^N \omega_i (-1)^{y_i} x_i$ where $\omega_i \geq 0$ for all $i = 1, \cdots, N$. Since $\langle \theta^*, (-1)^{y_i} x_i\rangle \geq 0$, this implies that $\langle \theta^*, x\rangle \geq 0$ and consequently the expert must have classified $x$ as 0. Likewise, for inputs $\in -\mathcal{K}$, $x$ can be expressed as a linear combination of $(-1)^{y_i} x_i$'s with non-positive coefficients. Consequently, the expert must have played the action 1 at these states.

(ii) We first prove that for any $x \notin \mathcal{C}$, one can find a $\theta' \in \Theta$ such that $\mathsf{sgn}(\langle \theta', x\rangle) \neq \mathsf{sgn}(\langle \theta^*, x\rangle)$. We subsequently prove that this result implies the required statement.

This result can be proved in two cases: (a) assuming that $\langle \theta^*, x\rangle \geq 0$ and (b) $\langle \theta^*, x\rangle < 0$. We prove the case (a) and argue that (b) follows similarly. For any input $x$ such that $\langle \theta^*, x\rangle \geq 0$, we show that there exists a $\theta \in \mathcal{K}$ such that $\langle \theta, x\rangle < 0$.

Indeed, in the first case, for any point $v \in \mathcal{K}$, the inner product $\langle v, \theta\rangle \geq 0$ for all $\theta \in \mathcal{K}$. This is because for every $\theta \in \mathcal{K}$, $\langle \theta, (-1)^{y_i} x_i\rangle \geq 0$ and every point $v$ in $\mathcal{K}$ is a positive linear combination of the $(-1)^{y_i} x$ vectors. Therefore if $v \notin \mathcal{C}$, there exists a $\theta \in \mathcal{K}$ such that $\langle v, \theta\rangle < 0$ and choosing $v = x$ completes the proof.

Finally, observe that from (i), and since $\theta^* \in \mathcal{K}$ (we can repeat the same argument for $-\mathcal{K}$), the cone $\mathcal{K}$ corresponds to $\cap_{\theta \in \Theta} \mathcal{H}_\theta$, where $\mathcal{H}_\theta$ is the halfspace $\{x \in \mathbb{R}^d : \langle x, \theta\rangle \geq 0\}$. For any $\widehat{\theta}$, by using Demorgan's laws it is easy to verify that this is a superset of $\left(\cup_{\theta \in \Theta} (\mathcal{V} \cap \mathcal{H}_\theta)^c\right)^c$ for any $\mathcal{V} \subseteq X$. In particular, choosing $\mathcal{V}$ as the set of inputs classified as 0 by $\widehat{h}$ (output classifier of the confidence set linear classification algorithm), we see that for every input $x \notin \mathcal{K}$, there exists a $\theta \in \Theta$ such that $x \in \mathcal{V}$ but $x \notin \mathcal{H}_\theta$ or vice versa. Namely the output predicted by $\widehat{h}$ on the point $x$ differs from the output predicted by some linear classifier $\theta \in \Theta$. $\qquad\square$

### 7.7 Proof of Theorem 7

**Theorem 12.** *Fix $d > 1$. Draw points $S_0, S_1, S_2, \ldots, S_{n+1}$ uniform random points on a half sphere in $\mathbb{R}^d$. Let $\mathcal{K}$ be the convex cone spanned by $S_1, \ldots, S_n$. Then there exists universal constants $c, C > 0$ such that*

$$\frac{cd^{3/2}}{n\sqrt{\log d}} \leq \mathbb{P}[S_0 \notin \mathcal{K}] \leq \frac{Cd^{3/2}\log d}{n} \tag{61}$$

*holds for sufficiently large $n$.*

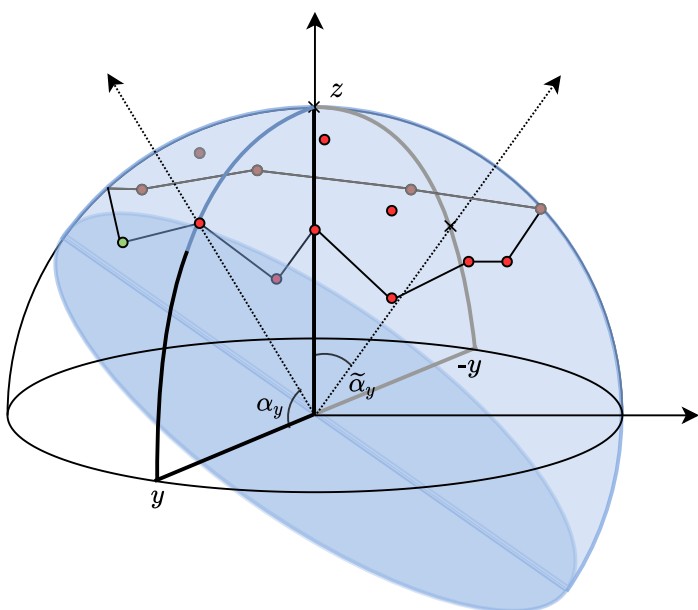

Figure 3: The red points represent $S_1, \cdots, S_n$. The green point $w \in \mathcal{K}$ satsifies $\langle z \cos \alpha - y \sin \alpha, w \rangle = 0$

*Proof.* First, observe that it is sufficient to prove the bound (61) (with possibly different constants $c, C$) when $S_1, \ldots, S_n$ are replaced by points in a Poisson process the half sphere with intensity $\frac{2n}{\mathsf{vol}(\mathbb{S}^{d-1})}$. Indeed, this can be show using the facts that $\mathbb{P}[S_0 \notin \mathcal{K}]$ is monotonically decreasing in $n$ and that there are more than $2n$ or fewer than $n/2$ Poisson points with exponentially small probability.

Henceforth, we shall consider the Poisson version of the problem. Note that the boundary of the half sphere is $\mathbb{S}^{d-2}$. Denote by $z$ the north pole of the hemisphere. For each $y \in \mathbb{S}^{d-2}$, define

$$\alpha_y := \sup\{\alpha \in [0, \pi/2] \colon y \cos \alpha + z \sin \alpha \notin \mathcal{K}\}. \tag{62}$$

Moreover, define

$$\tilde{\alpha}_y := \sup\{\alpha \in [0, \pi/2] \colon \langle z \cos \alpha - y \sin \alpha, w \rangle \geq 0, \ \forall w \in \mathcal{K}\}. \tag{63}$$

Then the Hahn-Banach theorem, we see that $\alpha_y$ is the smallest value such that

$$\langle y \cos \alpha_y + z \sin \alpha_y, -\tilde{y} \sin \tilde{\alpha}_{\tilde{y}} + z \cos \tilde{\alpha}_{\tilde{y}} \rangle \geq 0, \quad \forall \tilde{y}. \tag{64}$$

Equivalently,

$$\tan \alpha_y = \sup_{\tilde{y}} \langle y, \tilde{y} \rangle \tan \tilde{\alpha}_{\tilde{y}}. \tag{65}$$

The significance of $\alpha_y$ is the following: suppose $\kappa := \sup_y \tilde{\alpha}_y$. Then $\kappa = \sup_y \alpha_y$ in view of (65), and we have

$$\frac{2(1 - \cos^{d-1} \kappa)}{\mathsf{vol}(\mathbb{S}^{d-1})} \int_{y \in \mathbb{S}^{d-2}} \alpha_y dy \leq \mathbb{P}[S_0 \notin \mathcal{K}|\mathcal{K}] \leq \frac{2}{\mathsf{vol}(\mathbb{S}^{d-1})} \int_{y \in \mathbb{S}^{d-2}} \alpha_y dy. \tag{66}$$

Thus, as long as $\kappa$ is small, the problem is reduced to estimating $\int_{y \in \mathbb{S}^{d-2}} \alpha_y dy$. Note that there exists a constant $c_1 > 0$ (which is allowed to depend on $d$) and a vanishing sequence $\kappa_1, \kappa_2, \cdots \to 0$ such that with probability at least $1 - e^{-c_1 n}$, we have $\kappa < \kappa_n$. Indeed, this can be shown using by choosing an $\epsilon$-net on $\mathbb{S}^{d-2}$ with $\epsilon = 0.1$ (say), and arguing that with high probability each of $\epsilon$-neighborhood in the hemisphere of a point in the net includes a Poisson point; the details are omitted.

Conditioned on $\kappa < \kappa_n$, (65) implies

$$\alpha_y = (1 + o_{\kappa_n}(1)) \sup_{\tilde{y}} \langle y, \tilde{y} \rangle \tilde{\alpha}_{\tilde{y}} \tag{67}$$

where explicitly, $1 + o_{\kappa_n}(1)$ here is a number between $\sup_{0 < \alpha < \kappa_n} \frac{\tan \alpha}{\alpha}$ and its inverse. Then

$$\mathbb{P}[S_0 \notin \mathcal{K}] = \mathbb{P}[S_0 \notin \mathcal{K} | \kappa < \kappa_n] + O(e^{-c_1 n}) \tag{68}$$

$$= \frac{2(1 + o_{\kappa_n}(1))}{\mathsf{vol}(\mathbb{S}^{d-1})} \int_{y \in \mathbb{S}^{d-2}} \mathbb{E}[\alpha_y | \kappa < \kappa_n] dy + O(e^{-c_1 n}) \tag{69}$$

$$= \frac{2(1 + o_{\kappa_n}(1))}{\mathsf{vol}(\mathbb{S}^{d-1})} \int_{y \in \mathbb{S}^{d-2}} \mathbb{E}[\sup_{\tilde{y}} \langle y, \tilde{y} \rangle \, \tilde{\alpha}_{\tilde{y}} | \kappa < \kappa_n] dy + O(e^{-c_1 n}). \tag{70}$$

Therefore, the theorem is proved if we show that

$$\frac{cd^{1.5}}{n\sqrt{\log d}} \le \frac{1}{\mathsf{vol}(\mathbb{S}^{d-1})} \int_{y \in \mathbb{S}^{d-2}} \mathbb{E}[\sup_{\tilde{y}} \langle y, \tilde{y} \rangle \, \tilde{\alpha}_{\tilde{y}} | \kappa < \kappa_n] dy \le \frac{Cd^{1.5} \log d}{n} \tag{71}$$

for possibly different constants $c, C$.

Now let $\phi$ be the map from the hemisphere to $\mathbb{S}^{d-2} \times [0, \frac{\mathsf{vol}(\mathbb{S}^{d-1})}{2\mathsf{vol}(\mathbb{S}^{d-2})})$ which is measure preserving and satisfies $\phi_1(y \cos \alpha + z \sin \alpha) = y$ for any $y \in \mathbb{S}^{d-2}$ and $\alpha \in [0, \pi/2)$, where $\phi_1, \phi_2$ denote the first and the second coordinates of the value of $\phi$. We have

$$(\cos^{d-1} \alpha) \cdot \alpha \le \phi_2(y \cos \alpha + z \sin \alpha) \le \alpha. \tag{72}$$

Under $\phi$, the images of the Poisson points on the hemisphere still behave as a Poisson point process with intensity $\frac{2n}{\mathsf{vol}(\mathbb{S}^{d-1})}$.

Now define for each $y \in \mathbb{S}^{d-2}$,

$$\tilde{\theta}_y := \sup\{t \ge 0 : \text{ no } \phi\text{-image of Poisson point in } V(y, t)\} \tag{73}$$

where $V(y, t)$ is defined as the set of $(a, b) \in \mathbb{S}^{d-2} \times [0, \infty)$ such that $b < t \cos \angle(y, a)$. Define for each $x \in \mathbb{S}^{d-2}$,

$$\theta_x := \sup_{y \in \mathbb{S}^{d-2}} \tilde{\theta}_y \cos \angle(x, y). \tag{74}$$

Using linear algebra, we can show that given $\kappa < \kappa_n$, we have

$$\tilde{\theta}_y = (1 + o_{\kappa_n}(1))\tilde{\alpha}_n \tag{75}$$

Therefore, by the observation in (71), we prove the theorem if we show the following

$$\frac{cd^{1.5}}{n\sqrt{\log d}} \le \frac{1}{\mathsf{vol}(\mathbb{S}^{d-1})} \int_{y \in \mathbb{S}^{d-2}} \mathbb{E}[\sup_{\tilde{y}} \langle y, \tilde{y} \rangle \, \tilde{\theta}_{\tilde{y}} | \kappa < \kappa_n] dy \le \frac{Cd^{1.5} \log d}{n}. \tag{76}$$

By symmetry and $\mathsf{vol}(\mathbb{S}^{d-2})/\mathsf{vol}(\mathbb{S}^{d-1}) = \Theta(d^{1/2})$, the above is equivalent to

$$\frac{cd}{n\sqrt{\log d}} \le \mathbb{E}[\sup_{\tilde{y}} \langle y, \tilde{y} \rangle \, \tilde{\theta}_{\tilde{y}} | \kappa < \kappa_n] \le \frac{Cd \log d}{n}. \tag{77}$$

Since $\mathbb{P}[\kappa < \kappa_n] \ge 1 - e^{-c_1 n}$, it suffices to show the above bound with two modifications of the problem: 1) in the definition of $\tilde{\theta}_y$, the Poisson process is extended to one with the same density $\frac{2n}{\mathsf{vol}(\mathbb{S}^{d-1})}$ but on $\mathbb{S}^{d-2} \times [0, \infty)$; 2) uncondition on $\kappa < \kappa_n$, namely,

$$\frac{cd}{n\sqrt{\log d}} \le \mathbb{E}[\sup_{\tilde{y}} \langle y, \tilde{y} \rangle \, \tilde{\theta}_{\tilde{y}}] \le \frac{Cd \log d}{n}. \tag{78}$$

for possibly different universal constants $c, C > 0$. Now (78) is the content of Lemma 8; the difference in the factor $n$ is because we consider a Poisson process with intensity $\frac{2}{\mathsf{vol}(\mathbb{S}^{d-1})}$ instead in the lemma. The Theorem is now proved. $\qquad \square$

We shall establish the following lemma; the result for the original problem about random points on the half sphere will then easily follow. Note that a nice feature about the lemma is that there is no $N$ in the problem; the problem is purely depending on the dimension $d$.

**Lemma 8.** *Consider a Poisson point process on $\mathbb{S}^{d-2} \times [0, \infty)$ with intensity $\frac{2}{\text{vol}(\mathbb{S}^{d-1})}$. For each $y \in \mathbb{S}^{d-2}$, define*

$$\tilde{\theta}_y := \sup\{t \geq 0 : \text{ no Poisson point in } V(y, t)\} \tag{79}$$

*where $V(y, t)$ is defined as the set of $(a, b) \in \mathbb{S}^{d-2} \times [0, \infty)$ such that $b < t \cos \angle(y, a)$. Define for each $x \in \mathbb{S}^{d-2}$,*

$$\theta_x := \sup_{y \in \mathbb{S}^{d-2}} \tilde{\theta}_y \cos \angle(x, y). \tag{80}$$

*Then*

$$\frac{d}{\sqrt{\log d}} \lesssim \mathbb{E}[\theta_x] \lesssim d \log d. \tag{81}$$

*Proof.* We begin by first proving the upper bound.

$$\hat{\theta}_y := \sup\{t \geq 0 : \text{ no Poisson point in } \hat{V}(y, t)\} \tag{82}$$

where $\hat{V}(y, t)$ is defined as the set of $(a, b) \in \mathbb{S}^{d-2} \times [0, \infty)$ such that $b < t(\cos \angle(y, a))1\{\angle(y, a) < \frac{\pi}{2} - d^{-1/2}\}$.

Next, let $\mathcal{N}$ be an optimal $\epsilon$-covering of $\mathbb{S}^{d-2}$, where $\epsilon = \frac{1}{4}d^{-1/2}$. For each $y$ and $t$,

$$\mathbb{P}[\hat{\theta}_y > t] = \exp\left(-\text{vol}(\hat{V}(y, t)) \times \frac{2}{\text{vol}(\mathbb{S}^{d-1})}\right). \tag{83}$$

However,

$$\text{vol}(\hat{V}(y, t)) = \text{vol}((\cos d^{-1/2})B^{d-2}) = \cos^{d-2} d^{-1/2}\text{vol}(B^{d-2}). \tag{84}$$

Define $c_d := \cos^{d-2} d^{-1/2}$. Then

$$\mathbb{P}[\hat{\theta}_y > t] = \exp\left(-2tc_d \frac{\text{vol}(B^{d-2})}{\text{vol}(\mathbb{S}^{d-1})}\right) \tag{85}$$

$$= \exp\left(-\frac{2tc_d}{d-2} \frac{\text{vol}(\mathbb{S}^{d-3})}{\text{vol}(\mathbb{S}^{d-1})}\right) \tag{86}$$

$$= \exp(-\frac{tc_d}{\pi}). \tag{87}$$

Therefore by the union bound,

$$\mathbb{P}[\sup_{y \in \mathcal{N}} \hat{\theta}_y > t] \leq |\mathcal{N}|\exp(-\frac{tc_d}{\pi}) \tag{88}$$

$$\leq (3/\epsilon)^{d-2}\exp(-\frac{tc_d}{\pi}) \tag{89}$$

Let $T : \mathbb{S}^{d-2} \to \mathcal{N}$ be such that $\|T(y) - y\| \leq \epsilon$. Define $\hat{t} := \sup_{y \in \mathcal{N}} \hat{\theta}_y$. For any $y \in \mathbb{S}^{d-2}$, by the definition of $\hat{t}$, we can find a Poisson point $(a, b)$ such that

$$b \leq \hat{t} \cos \angle(T(y), a)1\{\angle(T(y), a) < \frac{\pi}{2} - d^{-1/2}\}. \tag{90}$$

Then, since

$$\frac{\cos \angle(T(y), a)1\{\cos \angle(T(y), a) < \frac{\pi}{2} - d^{-1/2}\}}{\cos \angle(y, a)} \leq \frac{\cos(\frac{\pi}{2} - d^{-1/2})}{\cos(\frac{\pi}{2} - d^{-1/2} + \epsilon')} \tag{91}$$

$$\leq \frac{\cos(\frac{\pi}{2} - d^{-1/2})}{\cos(\frac{\pi}{2} - \frac{1}{2}d^{-1/2})} \tag{92}$$

$$\leq 3 \tag{93}$$

for large $d$, where $\epsilon'$ denotes the angle between two points with distance $\epsilon$ on the sphere, we obtain

$$b \leq 3\hat{t}\cos\angle(y,a) \tag{94}$$

which implies that $\tilde{\theta}_y \leq 3\hat{t}$. Since this bound does not depend on $y$, we also obtain $\theta_x \leq 3\hat{t}$. This shows that

$$\mathbb{E}[\theta_x] \leq 3\mathbb{E}[\hat{t}] \leq 3\int_0^\infty \min\left\{(3/\epsilon)^{d-2}\exp(-\frac{t}{6\pi}),1\right\}dt \tag{95}$$

$$\lesssim \ln\left((3/\epsilon)^{d-2}\right) \tag{96}$$

$$\lesssim d\ln d \tag{97}$$

where we used $c_d \geq e^{-1/2} \geq 1/3$ for large $d$.

Now we move on to proving the lower bound. First, note that for any $t > 0$, the number of Poisson points in $\mathbb{S}^{d-2} \times [0,t]$ follows the Poisson distribution with strength bounded by:

$$\lambda = 2t\frac{\mathsf{vol}(\mathbb{S}^{d-2})}{\mathsf{vol}(\mathbb{S}^{d-1})} \tag{98}$$

$$\leq c\sqrt{d}t \tag{99}$$

where $c > 0$ is a universal constant. In particular, if

$$t = 0.1d(c\sqrt{d})^{-1}, \tag{100}$$

then the expected number of Poisson points is $\lambda = 0.1d$. By the Markov inequality, the number is smaller than $0.5d$ with probability at least $0.8$.

Let $\mathcal{E}_1$ be the event that there are at most $0.5d$ Poisson points in $\mathbb{S}^{d-2} \times [0,t]$, and denote by $\mathcal{P}_t$ the set of these Poisson points. Define

$$\mathcal{H} := \{h \in \mathbb{S}^{d-2}\colon \langle h,p\rangle \leq 0, \forall p \in \mathcal{P}_t\}. \tag{101}$$

We claim that there exists a universal constant $c_1 \in (0,\pi/2)$ such that $\mathcal{H}^{c_1}$ has normalized conic area at least $0.5$, with probability at least $0.5$ conditioned on $\mathcal{E}_1$; call this conditional event $\mathcal{E}_2$. Here, $\mathcal{H}^{c_1}$ is defined as the set of all points on $\mathbb{S}^{d-2}$ whose angle with some point in $\mathcal{H}$ is at most $c_1$.

In fact, the above claim can be proved if we show that the same claims holds when $\mathcal{H}$ is replaced by

$$\mathcal{H}' := \{h \in \mathbb{S}^{d-2}\colon \langle h,p\rangle \leq 0, \forall p \in \mathcal{B}\}. \tag{102}$$

where $\mathcal{B}$ is the set of $0.5d$ uniformly randomly drawn points $p_1,\ldots,p_{0.5d}$ on $\mathbb{S}^{d-2}$. Let $p_{0.5d+1},\ldots,p_{d-1}$ be an arbitrary basis of the orthogonal complement of the linear space spanned by $p_1,\ldots,p_{0.5d}$. With probability at least $0.5$, the nonzero singular values of the matrix $[p_1,\ldots,p_{0.5d}]$ are bounded in the interval $(c_2,c_3)$ where $c_2,c_3 > 0$ are universal constants; see e.g. https://djalil.chafai.net/docs/sing.pdf It follows that the singular values of $P := [p_1,p_2,\ldots,p_{d-1}]$ is bounded in the same interval. Note that if $r_1$ and $r_2$ are two rays with a small angle, then their linear transforms $Pr_1$ and $Pr_2$ satisfy $\frac{\angle(Pr_1,Pr_2)}{\angle(r_1,r_2)} \in (c_4,c_5)$, where $c_4,c_5 > 0$ are universal constants explicitly defined by $c_2,c_3$. It follows that under the linear transform, conic areas are changed by a factor within $(c_5^{-(d-2)},c_4^{-(d-2)})$. Then we note that the cone of $\mathcal{H}'$ equals $(P^\top)^{-1}$ applied to $\{s \in \mathbb{R}^{d-1}\colon \langle s,e_i\rangle \leq 0, i = 1,\ldots,0.5d\}$; indeed, for each $i$,

$$\{s\colon \langle s,p_i\rangle = 0\} = \{s\colon \langle s,Pe_i\rangle = 0\} \tag{103}$$

$$= \{s\colon \langle P^\top s,e_i\rangle = 0\} \tag{104}$$

$$= \{(P^\top)^{-1}s\colon \langle s,e_i\rangle = 0\}. \tag{105}$$

Therefore, the normalized conic area of $\mathcal{H}'$ is at least $c_5^{-(d-2)}2^{-0.5d}$. It then follows by the isoperimetry on the sphere that there exists $c_1 \in (0,\pi/2)$ such that $\mathcal{H}'^{c_1}$ has normalized conic area at least $0.5$. This proves the claim following (101).

Next, recall that $x$ is a fixed point on $\mathbb{S}^{d-2}$. Conditioned on $\mathcal{E}_1,\mathcal{E}_2$, by the rotation invariance, $x \in \mathcal{H}^{c_1}$ with probability at least $0.5$; call this conditional event $\mathcal{E}_3$. Under $\mathcal{E}_3$ Let $y$ be a point in $\mathcal{H}$ closest

to $x$. Then $\angle(x, y) \leq c_1$. Note that $\mathcal{E}_1, \mathcal{E}_2, \mathcal{E}_3$ are determined by the Poisson point configurations in $\mathbb{S}^{d-2} \times [0, t]$, which is independent of the configurations in $\mathbb{S}^{d-2} \times (t, \infty)$. Now set

$$t' := \frac{\sqrt{d}}{10\sqrt{\log d}} t. \tag{106}$$

We claim that conditioned on $\mathcal{E}_1, \mathcal{E}_2, \mathcal{E}_3$,

$$\theta_y \geq t' \tag{107}$$

with probability at least $0.9$; call this conditional event $\mathcal{E}_4$. Indeed, the above bound on $\theta_y$ is equivalent to the event that there is no Poisson point $(a, b)$ satisfying

$$b < t' \cos \angle(y, a). \tag{108}$$

Note that by the definition of $y$, there is no Poisson point for which $b \in [0, t)$. Therefore (conditioned on $\mathcal{E}_1, \mathcal{E}_2, \mathcal{E}_3$), (107) is equivalent to the event that there is no Poisson point in

$$\{(a, b) \colon b \in (t, t' \cos \angle(y, a))\}. \tag{109}$$

The Lebesgue measure of the above region equals $t'\mathsf{vol}(\sin\arccos\frac{t}{t'}B^{d-2})$, where $B^{d-2}$ is the unit ball. Therefore, no Poisson point in (109) with probability

$$\exp\left(-t'\mathsf{vol}(\sin\arccos\frac{t}{t'}B^{d-2}) \cdot \frac{2}{\mathsf{vol}(\mathbb{S}^{d-1})}\right)$$

$$= \exp\left(-\frac{t'}{\pi}(\sin\arccos\frac{t}{t'})^{d-2}\right) \tag{110}$$

$$\geq \exp\left(-\frac{t'}{\pi}\left(1 - \frac{100\log d}{d}\right)^{\frac{d-2}{2}}\right) \tag{111}$$

$$\geq 0.9 \tag{112}$$

for large enough $d$. This proves the claim around (107).

Finally, under all the events $\mathcal{E}_1, \mathcal{E}_2, \mathcal{E}_3, \mathcal{E}_4$, which happens with probability at least $0.1$, we have

$$\theta_x \geq \cos c_1 \theta_y \geq t' \cos c_1 \tag{113}$$

which is $\Theta(\frac{d}{\sqrt{\log d}})$, thus the lower bound is established. $\qquad\square$