# OpenReview forum: "On the Value of Interaction and Function Approximation in Imitation Learning"
_NeurIPS.cc/2021/Conference — NeurIPS 2021 Poster_

### Official Review · Reviewer_AQx2 · 2021-07-16

**Rating:** 7
**Confidence:** 4

**Summary:**

This paper investigates statistical guarantees for imitiation learning (IL) under the $\mu$-recoverability assumption of [27]. The authors prove  bounds on the sub-optimality of any algorithm which does not interact with the MDP and uses an offline dataset. Under linear function approximation, the authors prove bounds on the suboptimality of behaviour cloning.

**Limitations And Societal Impact:**

Yes.

**Main Review:**

Overall I found this paper to be quite good, for which I am recommending an accept.

I really appreciated the informal theorems presented before the real theorems, as they help in the clarity of the paper.

I think the proofs in the Appendix could be clarified and extended to make it easier for more readers to follow, as currently it seems to be written for an audience that is already quite familiar with that type of theoretical work.

More details below.

## Questions / Comments
1.  What exactly does "0-1 loss" mean in line 84?
1. In line 92, why does $\hat{\pi}$ denote the learner, if the policy is continuously changing? Relatedly, in line 93 does $J(\hat{\pi})$ represent the policy at convergence? Same question on line 103. The phrasing used in informal theorem 3 seems better for this.
1. Does the negative result of informal theorem 2 not hold for $\mu < 1$? Why not?
1.  The term "statistical minimax rate" is used quite a lot in the paper but never explicitly defined. It would be good to define for clarity.
1.   In line 149, "it is known from [22] that...", is this also under the assumption of deterministic transitions, like in the current paper?
1.  In line 168, if $h^{*}$ is defined as an $\arg\max$, shouldn't you use "$=$" instead of "$\sim$"?
1.  Please define $\mathbb{S}^{d-1}$ in line 188.
1.  In line 193, the term $d^{3/2} / N$ seems to suggest that there is a tradeoff between expressivity (large $d$) and vacuosity of bound (small $d$), since $d \ll |\mathcal{S}|$ likely incurs larger $\mathfrak{R}$. Is this the case?
1.  In line 230, shouldn't it be $\hat{\pi}^1,\cdots,\hat{\pi}^T$?
1.  In line 237, isn't the second term of the _empirical online-learning regret_ just zero?
1.  The statement in line 245: "By scaling each reward..." seems a little strange. Presumably $\mu$ has relevance under the assumption that the rewards are in $[0, 1]$ (as stated in the introduction). But just scaling the rewards, while not changing optimal policies, _does_ change the meaning and tightness of suboptimality bounds (e.g. I can make the bounds arbitrarily small by scaling the rewards). I guess this is for the lower bound, so it's ok? On this note, is $\mu$-recoverability necessary just so you satisfy the the existing theorem's requirements (from [22])?
1.  In Theorem 3, it seems the lower bound would be of more use if it depended on $|\mathcal{A}|$ instead of $H$.
1.  In line 322, I think you need to add "with probability $1-\delta$" to the end of the sentence.
1. In Section 4 you should specify from the beginning that you are considering the case when $P$ is known (this is only revealed once you reach the Algorithm).
1.  In Algorithm 1, it would help clarify things if you highlight with a different color what is different from the original MIMIC-MD.
1. In equation (8), what is $-\mathcal{K}$?
1.  In line 564 of the Appendix, it says "samples the policy $tr_i$", do you mean policy or trajectory?
1.  In line 565 it says "rolling out $tr_i$, should this be $\pi_i$?
1.  In line 568, what does it mean to be rolled out and conditioned on $\hat{\pi}^i$?


## Minor comments
1.  What do you mean by "nicely" in line 29?
1.  What do you mean by "game AI" in line 37?
1.  Throughout the paper the authors use numbered citations as nouns (e.g. "[25] study BC from a theoretical..."). You should use something like "Ross & Bagnell [25] study BC from a theoretical...". This is especially problematic when you start sentences/paragraphs like this, as you do in line 320.
1.  In line 81, in the expectation subscript, use $a'$ (instead of $a$) to avoid confusion, as $a$ already has a different meaning.
1.  Line 87, the sentence "the size of expert dataset / number of MDP interactions, N in the no-interaction / active settings." reads weirdly, consider rephrasing.
1.  Line 89, should be "propose **a** policy".
1.  In line 98 a sentence starts improperly ("idea is crucial...")
1.  In line 130 please state what ERM stands for.
1.  Line 146, should read "provied **a** dataset $D$".
1.  Line 164, should be "output" instead of "otput".
1.  Section 1.1 (Related work) seems out of place where it is, as it interrupts the narrative flow. Consider putting at the end of the paper.
1. Line 235, should be "a measurable function **of** the first".
1.  Line 282 should be "upper bound" (singular, not plural).
1.  Line 311, what do you mean by "By contradiction"?
1.  In the paper use \restatable so that the theorem statements are reprinted in the Appendix.

**Time Spent Reviewing:**

4

---

> ### Author Response · Authors · 2021-08-10
> **Response to reviewer AQx2**
>
> We thank the reviewer for the feedback.
> We would like to clarify in the summary provided by the reviewer that we also prove bounds in the linear setting when the MDP transition is known. Here it is possible to improve on the guarantees of behavior cloning by reducing IL to the problem of “confidence set classification problem”. We address the comments provided by the reviewer in sequence below:
>
> 1. The $0$-$1$ loss here refers to the probability that the learner plays an action different from the expert, when the underlying state is generated by rolling out the learner's policy. Namely, $\frac{1}{H} \sum_{t=1}^H \mathbb{E}_{\hat{\pi}} [ \mathbb{E}\_{a_t \sim \pi_t (\cdot|s\_t)} [ \mathbb{I} (a_t \ne \pi^*\_t (s\_t))] ]$.
>
> 2. The notation $\widehat{\pi}$ is used to define the learner policy at convergence. There is no criterion that it must be defined by an iterative algorithm per se. It refers to the final policy returned by the learner using some algorithm.
>
> 3. For any value of $\mu$, note that the error incurred by any policy compared to the expert policy is upper bounded by $H \mu$. This can be seen by writing the $J(\pi^*) - J(\pi)$ for any policy $\pi$ as a cascading sum: $\sum_{k=1}^H J(\pi^k) - J(\pi^{k-1})$ where $\pi^k$ plays the policy $\pi^*$ until (and including) time $k$ and subsequently plays actions according to the policy $\widehat{\pi}$. By expanding using their definitions and using the recoverability property, for each $k$, the term $J(\pi^k) - J(\pi^{k-1})$ can be upper bounded by $\mu$. If $\mu$ is extremely small $\ll 1$, then $H \mu$ becomes extremely small. In the limit $\mu \to 0$, all policies nearly achieve the same value on the IL instance (so an arbitrary policy is near-optimal here). The point is that the case $\mu \to 0$ is vacuous and any lower bound only makes sense when it is not arbitrarily small. In fact, our lower bound result is in fact exactly $\min \{ SH^2/N, \mu H, H\}$ and we ignore the second term by assuming $\mu > 1$. We shall include this in the statement of the main result if the reviewer feels so appropriate.
>
> 4. The term "minimax rate" is standard terminology in the statistics and the decision theory literature. In defining this notation, we would be at risk of being superfluous since these definitions are fairly mainstream and accessible in most standard statistics textbooks.
>
> 5. Correction: The current paper assumes a *deterministic expert policy*, but not deterministic transitions. The cited paper [22] also applies under the same setting.
>
> 6. Thank you for catching this typo. We will correct the $\tilde{}$ to $=$.
>
> 7. $\mathbb{S}^{d-1}$ denotes the surface of the unit $L_2$ sphere in $d$ dimensions. We shall include a formal definition where it first appears.
>
> 8. We believe this should be the case in both theory and practice. Indeed when the features are made higher dimensional, many more policies can be realized in the linear-expert setting. By extension, identifying which of these policies is the expert is now a more challenging statistical problem. In practice there would be no reason to go beyond the smallest set of features with enough expressivity to model the expert behavior to a sufficient degree of accuracy.
>
> 9. Isn't that what appears in the paper?
>
> 10. That is correct. We simply write this term to relate to the notion of regret-in-hindsight considered in online learning where the sequence of iterates output by the learner is compared with the single best choice in hindsight (which incurs $0$regret here).
>
> 11. The lower bound instances considered in [22] do not satisfy $\mu$-recoverability and are incredibly challenging in the sense that a single mistake is catastrophic (and incurs a suboptimality of $H$). Scaling down the rewards by $\mu/H$ may seem strange, but this allows mistakes to be less catastrophic - now they only incur an error of at most $\mu/H \times H = \mu$. This is the intuition for the lower bound under $\mu$-recoverability.
>
> 12. The lower bound can be expressed in terms of $|\mathcal{A}|$ and $H$. While it is not explicitly given in the appendix, the guarantee can easily be gleaned from Lemmas 6 and 7 and is of the order $\min \{ 1, SH/N \} \times \left(1 - \frac{1}{|\mathcal{A}|} \right)^{H+1}$. The case when $|\mathcal{A}| \ll H$, however, is not very interesting as the lower bound becomes exponentially small in $H$ as $|\mathcal{A}|$ approaches a constant. In this regime, it is necessary to construct a different lower bound instance.
>
> 13. We will add a quantifier to this effect to the statement here.
>
> 14. In section 4 in the title (as well as in the first paragraph), the known-transition setting is mentioned. Definition 6 in the introduction mentions the precise setting here (knowledge of $P$ and initial state distribution).
>
> 15. The original Mimic-MD is defined for the tabular setting. The current algorithm modifies several steps of mimic-MD for the linear setting. Rather than highlighting the differences, we propose to include a description of mimic-MD for the tabular setting in the Appendix in case a reader wishes to refer to it. We shall also include a short remark in the paper clarifying the differences.
>
> 16. For a set $\mathcal{K} \subseteq \mathbb{R}^d$, $ - \mathcal{K}$ is defined as $ \{ -x : x \in \mathcal{K} \}$. In other words, it is the mirror image of the set about the origin. This is a standard notation in geometry, but we can include a definition here to avoid any ambiguity.
>
> 17, 18, 19. The statement here should read: "... samples the policy $\pi_i$ from some distribution conditioned on $\text{tr}\_1 , \cdots , \text{tr}\_{i-1}$, subsequently samples a trajectory $\text{tr}\_i$ by rolling out $\text{tr}\_i$ and repeats for $N$ iterations. Line 568 should be interpreted as two sentences: "The trajectory $\text{tr}\_i$ is rolled out using $\pi\_i$. Conditioned on $\pi\_i$, ...". We hope this resolves the ambiguity here.
>
> Below we address some of the minor comments (which have questions). We will resolve the remaining in the subsequent version of the paper.
>
> 1. "Nicely" here just means to whatever specification the task requires - such as driving a car safely.
>
> 2. We use the term game AI to refer to autonomous game playing agents for games such as (famously) Chess, Go, Starcraft and DOTA 2 among others.
>
> 8. ERM is an abbreviation for empirical risk minimizer. We will include the full form in the final version.
>
> 14. "By contradiction" is short for "Proof by contradiction" which is a standard tool in logic (proof by assuming the opposite).

---

> > ### Comment · Reviewer_AQx2 · 2021-08-16
> > **Score unchanged**
> >
> > Thank you for the clarifications, and my score remains unchanged at a 7 :).

---

### Official Review · Reviewer_A7G7 · 2021-07-16

**Rating:** 7
**Confidence:** 3

**Summary:**

They study imitation learning in several variations, in particular with interaction (i.e., the expert can be queried during training) and without. And under some interesting additional assumptions, such as mu-recoverability (i.e., if the imitator visits suboptimal states, this has small impact of the overall value, by being able to go back to optimal easily).

They provide several theoretical results,
- negative ones -- i.e., lower bounds on the suboptimality that can be achieved in terms of difference of imitator versus optimal expert value incurred -- as well as
- positive ones -- algorithms that achieve linear instead of quadratic suboptimality, in particular their main contribution thm4. This works by reduction to supervised learning paradigm sample bounds.

As a means to this, they also have a little taxonomy on versions of IL.

One thing that's hard to judge for me is novelty. I feel this is novel, but I'm not sure how much overlap there is with DAgger and all the subsequent work in this direction, not an expert on this.

**Limitations And Societal Impact:**

Small limitations include:
- I'm surprised that one of the most common settings -- no interaction, but being able to sample from the dynamics -- is not discussed. However, I guess Def6 is a close enough approximation to this for the sake of this paper.
- How limiting is the linear expert setting? Linear would be pretty limiting, but I guess it's only linear in the features which would be quite general again.
- There are no experiments but that's OK given there is enough conceptual and theoretical contribution.
- I have not read the full paper. Some writing may be premature -- e.g. l98 "regard. idea is crucial" -- but this seems neglectable.

**Main Review:**

Overall this is an interesting paper, addressing common IL settings, providing non-trivial theoretical contributions.

The reduction-based approach to thm4 looks powerful (though reduction, I guess, are not new for this but originate from DAgger etc.).

The little IL taxonomy (Def1, 2, ...) is interesting in itself.

Some of the proofs look non-trivial.

Quality-wise, I have not checked details, but the theorems and proofs seem to be stated with care.

Style of writing is OK-good.

**Time Spent Reviewing:**

3

---

> ### Author Response · Authors · 2021-08-10
> **Response to Reviewer A7G7**
>
> We thank the reviewer for the comments and will correct the typographic errors in the subsequent version of the paper.
> Addressing the two main questions:
>
> 1. In this paper we leave the study of the setting where the learner can interact with the MDP for a finite number of trajectories for future work and focus on the known transition setting. There are two reasons for this choice:
>
> (i) in arguably one of the most challenging settings for practical reinforcement learning today, autonomous driving, a huge fraction of algorithm design and tuning can be attributed to learning in an artificial simulation environment. In support of this statement, Waymo's self driving workflow relies heavily on a simulated environment which can be used to augment a single experience collected on public roads with millions of variations such as with cars or pedestrians on the road. In the absence of this feature, it would be impossible to deploy algorithms directly in practice and hope for meaningful results. More information is available in the public domain here: https://ltad.com/about/simulation.html
>
> (ii) A more nuanced point is that for many problems in RL (eg. planning), algorithms in the known transition setting (say, value iteration (VI) ) are often the stepping stones to construct good algorithms in the presence of online exploration (namely, UCB-VI). We take the same approach here by proposing optimal algorithms for the known transition setting. In the tabular setting, there has already been some recent work addressing the problem in the online exploration setting in [Xu21]. Extending these approaches to the linear setting which we consider is still open.
>
> 2. We should mention that the linear assumption we study here is weaker than the linear function approximation setting studied in practice. Indeed, we only require that the expert policy can be “realized” by a linear classifier - this does not require the optimal $Q$ function ($Q^*$) to be linear (which is the typical assumption). Notwithstanding this point, the linear setting is the correct stepping stone to extend these results to even more challenging function approximation settings, such as the cases of bounded Eluder dimension and bounded Bellman rank. By providing algorithmic insights in the linear setting (which when specialized to the tabular setting are in fact optimal), we believe that there is scope to extend the resulting guarantees to cases with general function approximation.
>
> [Xu21]: “Nearly Minimax Optimal Adversarial Imitation Learning with Known and Unknown Transitions”, T. Xu, Z. Li, Y. Yu,

---

> > ### Comment · Reviewer_A7G7 · 2021-08-30
> > **Thanks for the responses.**
> >
> > -

---

### Official Review · Reviewer_wRYy · 2021-07-17

**Rating:** 7
**Confidence:** 2

**Summary:**

 The authors study statistical guarantees for the Imitation Learning problem, for many settings: considering or not the possibility of interacting with the environment, using the \mu-recoverability assumption, and assuming linear function approximation. Furthermore, they introduce a new setting, called confidence set linear classification.

**Limitations And Societal Impact:**

No potential negative impact is present.

**Main Review:**

## Originality:
The main novelty consists in the analysis proposed by the authors, which allows to determine theoretical properties, improving some pre-existent results. Moreover, they introduce and study a new setting.

## Quality:
The applied methodology seems to be sound, and the work is complete, analysing a spectrum of interesting cases.

## Clarity:
While the paper is clearly well written, the structure of the whole work can be improved. The authors tried to summarize in the introduction the main results provided, by means of informal theorems and definitions. I would suggest to keep a more traditional informal introduction, in which formulas and statement are few, and the main ideas are explained by words. Consequently, I would suggest to add a preliminaries section, where both the notation part and the definition could fit more.

## Significance:
The work offers important statistical guarantees for different settings, advancing the state of the art. The obtained results are likely to be useful for other researchers to build on them.

**Time Spent Reviewing:**

2

---

> ### Author Response · Authors · 2021-08-10
> **Response to Reviewer wRYy**
>
> We thank the reviewer for the positive feedback!
> Space permitting, we shall modify the introduction to be less terse and include a preliminaries section with the formal mathematical definitions.

---

### Official Review · Reviewer_VnQb · 2021-07-19

**Rating:** 7
**Confidence:** 4

**Summary:**

The paper examines statistical guarantees of imitation learning under various regimes -- interactive setting, non-interactive setting, and non-interactive setting with known dynamics. It further looks at the linear setting and derives bounds for non-interactive settings with/without known dynamics. The main contributions are:
1. Interactive: To show that “reduction-based” online algorithms (DAGGER / AGGREVATE, DAEQUIL) have matching upper and lower bounds
2. Non-interactive: All such algorithms have O(H^2) bounds
3. Non-interactive + Linear + Parameter sharing: Recover O(H) bound
4. Non-interactive + Known dynamics + Linear: Introduces a new algorithm, confidence set linear classification, that achieves O(H^3/2)


**Limitations And Societal Impact:**

Yes

**Main Review:**

The paper addresses an important, yet under examined, problem -- what is the best one can expect from any imitation learning algorithm under various settings? The contributions in examining the linear setting are particularly important, one hopes functional analysis (RKHS) soon follows.

I very much enjoyed reading the paper. It made central ideas accessible to the reader. I much appreciated the effort put in the introduction to give the reader a broad view of the key results.

The two main strengths of the paper are:
1. Technically sound [4/5]: While some of the proofs get a bit hairy, there is adequate hand-holding and it is clear that the authors wield technical theorems in an adept manner
2. Significance of claims (if true) [4/5]: In particular, the linear analysis potentially paves the way for a full RKHS analysis and that would be truly consequential.

However, there are two main shortcomings that keep the paper from being a strong addition to the body of imitation learning knowledge:
1. Contented contributions [2/5]: There appears to be contributions that are either incorrect or misinterpreted, and others that are previously established and refined here. Clarifying / removing these would sharpen the contributions this paper has to offer.
2. Clarity of exposition [3/5]: For practitioners in IL (which is still a large majority), this paper is hard to read and glean insights from. Insights are often lost in the paper and a concern is that the consequences of the theory presented in this paper may be lost on its readers. \

I will now expand on the two shortcomings

## Contented contributions
Before going into individual contributions, I would like to flag that there is a recent paper that appear to make a subset of the claims presented here that the authors do not cite:

Swamy et al. “Of Moments and Matching: A Game-Theoretic Framework for Closing the Imitation Gap” (ICML 2021) https://arxiv.org/abs/2103.03236

For instance, Swamy et al. also have a notion of \mu-recoverability (Def 3) and express bounds as a function of \mu. Hence, for overlapping claims (some of which I will attempt to call out), it would be useful to clarify the differences between them. I will henceforth refer to this paper as Swamy et al.

1. Interactive setting (matching upper and lower bounds) [Informal Theorem 1, Formal theorem 2,3]:
The upper bound is similar in form to Swamy et al. (as well as Dagger), i.e. \mu H \epsilon, where \epsilon is the per-step loss (I would imagine that would be of the order of |S| / N). The lower bound appears to be sharper than Swamy et al. (H \epsilon), and perhaps this is worth calling out.

The devices used in the upper bound proof also follow from reduction of imitation learning to online learning (learner provides a policy, receives a loss). Hence, it is unclear what this theorem adds to what is already established.

2. Non-interactive setting (Informal Theorem 1, Formal Theorem 2) The O(H^2) is a well known result and the proof uses an MDP very similar to the one used in “Efficient Reductions for Imitaiton Learning”  (http://proceedings.mlr.press/v9/ross10a/ross10aSupple.pdf), notably Figure 1.

3.  L.105, the authors mention “this is the first result to establish a clear separation in statistical minimax rate of suboptimality..”. This claim is not substantiated, and at the very least requires a clear comparison to Swamy et al. which does not make such a claim but conveys results of the same kind.

4. Non-interactive setting + Linear + Parameter sharing (Informal Theorem 4, Formal Theorem 5): The claim that parameter sharing results in a reduction from O(H^2) to O(H) appears to be incorrect. Notably in Lemma 1, \gamma can be very high which would make the bound in Lemma 1 H\gamma ~= H, i.e. trivial.

Taking a step back, it’s totally common to share parameters in behavior cloning (i.e. have the same weights for all timestep). The update in (4) would add up the errors over timestep and be exactly the supervised learning loss. \gamma can be high, and in fact be masking a H term, i.e. gamma = H \epsilon.

5. Non-interactive + Linear + Known dynamics (Informal Theorem 5, Formal Theorem 6)

Swamy et al. show that in this setting it is possible to get O(H \epsilon) though one needs a “planner” (ala MaxEnt IRL Ziebart et al.) to search over the sequence of actions. I was surprised to see no such discussion in this theorem although Mimic-MD clearly requires such a planner in step 6. Moreover, while O(H^3/2) seems to be a clear improvement over BC, O(H), it’s still not the same order as the bound O(H). Clarifying the significance of this result w.r.t to Swamy et al. would be useful.

6. Informal theorem 6 (formal theorem 7): This indeed appears to be the novel contribution. I struggled to follow the proof entirely but trust that the authors got it correct.


## Clarity of exposition
I’ll go over the paper chronologically and identify areas where the writing was unclear / there were typos.

Section 1: Introduction

* Should have been at most 2 pages. There was no need to have so many definitions up front. Certainly no need to even talk about linear results. Instead, I would have structured it to talk about the 3 regimes of IL, talk about the key results in each regime and have forward pointers to theorem in the paper. There is no need to state things twice.
* I would massively compress Def 4 onwards till the end. The only thing interesting there is the O(H) from parameter sharing and O(H^3/2) for known dynamics.
* I would keep the parts that  convey insight (L.59-66, L.73-79,L.155-165).
* L86. - 90: Unclear why reduction based approach fails to distinguish between power of learners. The authors claim they circumvent reduction based guarantees, but they don’t in their proofs (reduce it to FTRL).
* L.131: Classic supervised learning in IL does indeed use multi-class classification algorithm (for discrete actions > 2).

Section 2/3

* Move the definition here
* Theorem 1: Explicitly mention realizability assumption here.
* Theorem 3: Probably want the figure from appendix to be moved up here.
* L307: should be \theta^* without the _t subscript
* L310-313: Unclear what is meant by separable and Markovian here. The point is unclear as well.
* L316: The reader is still left trying to understand why, by mere rephrasing of the problem, the bound goes from O(H^2) to O(H). It feels like a smoke and mirror trick. The intuition is that to get such a reduction, surely we are paying elsewhere for added complexity (reasoning in the space of trajectories). That needs to be made clearer.

Section 4


L333: OPT: \tilde(J)_r(\pi^*)
Also one needs to explain how to solve OPT in practice and that it is hard.

Also, what if the reward function is only a function of state and not state action. Does this entire section fall apart? (Presumably no, but unclear)

Eq (5) explain E^c.
L340: Simple empirical estimate -- unclear from what distribution. Please explain mathematically

Alg 1, Line 6. \tilde(J)_r(\pi^*)

Appendix:
L565 tr_i should be \pi_i

L604: Wrong theorem (state the active setting)


**Time Spent Reviewing:**

48

---

> ### Author Response · Authors · 2021-08-10
> **Response to Reviewer VnQb**
>
> We thank the reviewer for the extremely detailed comments and appreciate the time taken to review the paper at this level of detail. We will try to incorporate changes in the subsequent version of the paper (such as a comparison with Swamy et al) and improving the clarity of the paper in the Appendix. Below we address the points from the contended contributions section sequentially:
>
> 1. The upper bound of $\mu H \epsilon$ of Ross et al reducing IL to online learning is precisely what we invoke here. However, the particular function of the number of trajectories “N” to which epsilon can be minimized is not clear from this result. Our contribution here is to show that there exists an online learning algorithm which in fact achieves epsilon scaling as $S/N$. This is not trivial as the online learning loss considered in the reduction of Ross et al is not a convex function (let alone smooth or strongly convex) in the variable parameter $\pi_i$, so existing algorithms/guarantees from the online learning literature cannot be invoked to prove this bound. Our contribution here is to show in spite of this, online mirror descent with entropy regularization can indeed minimize $\epsilon$ here to to $S/N$ which is optimal.
>
> 2. We argue that the lower bound in the non-interactive setting has quite a different structure from the lower bound in the one in “Efficient Reductions for IL”. As is pointed out in Rajaraman et al (2020), there is an algorithm for the latter instance which *achieves 0 suboptimality* given just a “single trajectory” from the expert policy. The argument in Ross et al (2010), is a lower bound showing that among the policies which approximately minimize 0-1 loss, there exists one which makes a fatal mistake and thus its error scales with $H^2$. It does not address whether every algorithm is forced to incur the $H^2$ dependence. In contrast, the lower bound of Rajaraman et al (2020) shows that, every algorithm must incur this $H^2$ dependence given a finite amount of expert data. Zooming into the particular instance, there is also quite a different structure here. The instance in Ross et al does not have the notion of a “bad state” where *every* learner is forced to be stuck at with decent probability (given a finite amount of expert data). In summary, the instance in Ross et al is not an information theoretic lower bound for the problem. Nevertheless, this is not a contribution in our current paper, and we use a significantly different approach to prove a lower bound in the known-transition setting by reducing the problem to mean estimation of binomial random variable given subsampled observations.
>
> 3. To the best of our knowledge, the results of Swamy et al do not prove any statements about the worst-case (i.e. minimax) sample complexity for IL and instead provide reduction based guarantees. While these are relevant in practice, this is not tantamount to saying that every algorithm has a bottleneck in achieving, say, $H^2$ dependence in the no-interaction setting. More importantly, we substantiate the claim of separating the sample complexity in the no-interaction and active settings by the means of Theorems 1 and 3. Theorem 1 shows that in the active setting, an error bound of $\mu SH/N$ can be achieved by “interactive” algorithms such as DAGGER. Theorem 3 shows that, in contrast, *even in the presence of mu-recoverability*, a suboptimality of $\Omega( S H^2 / N)$ is a worst-case lower bound for *every* algorithm which does not interact with the MDP (such as Behavior Cloning).
>
> 4. In general, *without* parameter sharing, the assertion is correct that the value $\gamma$ must depend on the horizon length $H$ as the reviewer points out. However, we show that with parameter sharing, the bound on $\gamma$ necessarily *does not* scale with $H$. Indeed, as we claim in Informal Theorem 4 / Theorem 5, $\gamma$ can be bounded by $d/N \log(\cdots)$ with parameter sharing. The intuition is as follows:  with parameter sharing, the learner can aggregate information across time steps in an episode to compute a single linear classifier
> $\hat{theta}$ with improved guarantees. Indeed, the amount of training data the learner has access to is now larger by a factor of $H$ (i.e. each trajectory provides $H$ samples of data for learning a single classifier common across time). This effectively reduces the dependence of $\gamma$ by a factor of $H$. In the reduction view, this kind of fine grained information is missing as one would be encouraged to abstract away $\gamma$ as $H \epsilon$. This does not capture the fact that epsilon is effectively much smaller (by a factor of $H$) when parameters are shared by virtue of the data aggregation across time. The concrete bound on $\gamma$ as $d/N \log (\cdots)$ captures this fact.
>
> 5. While the contribution in Swamy et al is correct, the question here is how small can $\epsilon$ be made as a function of the size of the expert dataset, $N$. Without this comparison, it is impossible to compare two algorithms as the respective “$\epsilon$’s” which can be achieved are different. For example in the presence of a planner, it is trivial (using $L_1$ distribution matching) to achieve a suboptimality of $H \sqrt{S/N}$. At first glance it seems easy to think that this has linear dependence on the horizon H. But the “sample complexity” here scales as  $N = SH^2 /epsilon^2$ which grows quadratically with H. In contrast, the sample complexity of Mimic-MD is $S H^{3/2} / \epsilon$ which improves on the dependence on $H$ as well as $\epsilon$. Secondly, and perhaps more importantly, the work of Rajaraman et al (2021) shows that in the worst case, *every* learner, even in the simplest tabular setting, must incur an error scaling as $H^{3/2} / N$ even if the transition is known perfectly. This means that the upper bound of $O (H \epsilon)$ proved by Swamy et al must have dependencies on $H$ present within $\epsilon$ (lest it contradict the established statistical lower bound).
>
>
> 6. Comments in Section 1: We will try to cut down on the introduction in a manner which preserves the key intuitions behind our contributions.
> In Line 86-90, we should mention the following: reductions are not algorithms. Reductions are a means to prove that algorithms achieve certain guarantees. Our main contribution is to “resolve” the reduction and establish a statistical guarantee for the online learning problem in the reduction. In conjunction with the $\mu$-recoverability reduction, this implies a statistical guarantee for the resulting algorithm which turns out to be optimal. Without “resolving” the reduction, the guarantee in Ross and Bagnell (2011) appears in terms of the loss in minimizing the online learning surrogate objective, and it is not clear a-priori how small it can be minimized to, given a finite amount of training data / interaction. For example, if it turned out that an exponential amount of data was required to make the surrogate objective sufficiently small (say a small constant), the reduction would not be very useful. This is not to say that the algorithm posited by a reduction guarantee is not good; it just means that the reduction *upper bound* does not give the full picture. Moreover, such a reduction (with exponential dependence in the sample complexity of the surrogate objective) also does not conclusively show that there is a fundamental difficulty in solving the imitation learning problem as there may be a *different proof* showing that the algorithm is in fact very sample efficient.
>
> 7. Comments in Section 2/3: We use the term Markovian policy per its standard RL / control theory definition, where actions played only depend on the current state and not the entire history of states visited until this time. In the language of optimization, a separable objective function is one where the optimization can be independently carried out over different variables. In this case, the optimization over the sequence of actions $\{ a_1,\cdots,a_H \}$ can be carried out over the variables $a_1,\cdots,a_H$ separately. The point here is that although, seemingly the learner chooses a sequence of actions for each distinct trajectory in the MDP (see eq. (4)), the resultant policy is indeed Markovian (i.e. the action played at a state depends only on the state and visited time).
>
> The argument in Line 316 is precisely what we illustrated above (in point (4)). The idea is that with parameter sharing, the learner has access to the same amount of training data to train a single classifier (compared to $H$ independent classifiers without parameter sharing). The effective increase in the size of the dataset enables the learner to carry out IL much more sample efficiently. This is the expected behavior for such a problem, and it is surprising that it has not appeared in more detail in the literature considering that (as the reviewers hints at too) parameter sharing is often carried out in practice for enabling sample and computational efficiency.
>
> 8. Comments in Section 4: The minimax optimization problem (OPT) was originally proposed in Rajaraman et al (2021) - there it is shown that in the presence of a simulator, OPT can be efficiently formulated as a convex optimization problem which can be solved efficiently. To address the reviewers point, the setting where the reward function is only a function of the state is a special case of state-action dependent reward setting, namely, by setting the reward for each action ‘a’ the same at every state ’s’. Therefore, the upper bounds continue to hold true (as this is now a special case of the s-a reward setting).
> We will clarify the definition of $\mathcal{E}^c$ in words to make it more intuitively clear what is going on here. We shall also clarify the term “simple empirical estimate" - the mathematical definition is concretely provided in the box for Algorithm 1, as being an empirical estimate for the event computed using the dataset $D_2$.

---

> > ### Comment · Reviewer_VnQb · 2021-08-18
> > **Clarified several misunderstandings!**
> >
> > I thank the authors for taking the time to provide such a thoughtful response! It was a joy to read and ponder.
> >
> > 4. I’ll start with my fundamental misunderstanding in the parameter sharing discussion: I forgot that we were in the realizable setting (at least temporarily forgot!) Once that is mentioned, it’s clear to see that epsilon can be driven down by just getting more data. So all things being equal, if you get H times more data, you get an O(H) reduction. Anyways, the clear phrasing in the rebuttal was very helpful in course correcting and perhaps it can make it back to the paper in some form.
> >
> > 1. Understood, the contribution is to show the dependence of epsilon on N, which none of the papers I referred to even attempt. I agree this is an important contribution
> >
> > 2. Looking at the proof again, I see the cleverness of the author’s MDP vs Ross and Bagnell MDP. I do think there is a notion of the bad state (s2 in RossMDP and b in the authorMDP). But while RossMDP has just 2 good states, the author introduces several good states, not all of which are visited. And since they are not visited, the learner can make a mistake and end up in the bad state. I think the proof is enlightening!
> >
> > 3. Yeah I see that Swamy et al do not prove any sample complexity bound, so it was unfair of me to say that they establish mini-max rate. I agree with the authors that their contributions here are novel.
> >
> > 5. The rebuttal point here was very helpful. Yes, one may get deceived into minimizing TV with their planner only to suffer worse sample complexity bounds H^2. MIMIC-MD indeed achieves better (still need to build intuition why). In fact, it achieves the minimax optimal rate so we can’t do better. Either ways, I agree this is an important contribution.
> >
> > 6. I loved this phrasing! In very lay terms, there is the Ross reduction is not tight enough to back out the sharpest bounds on the Ross algorithm. Perhaps the authors can add this phrasing in the appendix.
> >
> > 7, 8. Thanks for the clarifications!
> >
> > Based on my renewed understanding, I am more than happy to change my score to 7. I thank the authors once again for patiently enlightening this reviewer.

---

### Decision · Program_Chairs · 2021-09-27

**Decision:**

Accept (Poster)

**Comment:**

This is a good paper that studies the statistical properties of imitation learning under various regimes, e.g., interactive, and several non-interactive settings. The reviewers found the paper technically sound and the results significant.

The shortcomings of the papers are:
- Not citing and comparing with a recent paper by Swamy et al., ICML, 2021. Some of the results of these two papers are comparable.
I believe this omission in the initial submission is acceptable, given the recency of the paper and that its arXiv version was uploaded on 4 March 2021, relatively close to the NeurIPS deadline. That being said, I encourage the authors to provide a detailed comparison with that work.
- The exposition can be improved, for example, how the Introduction is presented.
Please consult the reviews for concrete suggestions.

Overall, I believe this paper should be accepted.